# Unlearning Incentivizes Learning under Privacy Risk

## Abstract

While machine learning empowers intelligent services and offers users customized experiences, privacy concerns emerge from regulatory requirements and the privacy-conscious demands of users. Machine unlearning presents a potential solution to these concerns. Despite the growing demand for practical deployment due to *the right to be forgotten* privacy regulations, the economic impact of machine unlearning on user behavior and platform profitability remains largely unexplored and may limit its implementation. In this paper, we formulate a set of contract design problems under both unlearning-disabled and unlearning-enabled scenarios. Challenges arise when the unlearning-enabled platform jointly designs compensation for both learning and unlearning to incentivize users' sequential decisions to balance the expected revenue and unlearning cost. We first conduct a questionnaire survey that reveals that machine unlearning increases users' willingness to participate in federated learning. We then provide a necessary condition for maximizing the surplus of an unlearning-enabled platform, enabling the point-wise decomposition for the optimal contract design problem, based on which we minimize the incentive cost and maximize the surplus for the platform. Our further analysis reveals that i) the incentive effects of unlearning grow quadratically with users' privacy sensitivity, and ii) enabling unlearning may even profit more than disabling it, under higher cost elasticity of risk distribution. Our numerical results show that the platform's profitability is primarily influenced by users' privacy sensitivity. When users are relatively highly privacy-sensitive, enabling unlearning can significantly improve profitability.

## CCS Concepts

• **Security and privacy → Economics of security and privacy**; • **Computing methodologies → Model development and analysis**.

## Keywords

Machine Unlearning, Contract Design, Risk Aversion

**ACM Reference Format:**
Anonymous Author(s). 2025. Unlearning Incentivizes Learning under Privacy Risk. In *Proceedings of The International World Wide Web Conference (WWW '2025)*. ACM, New York, NY, USA, 18 pages. https://doi.org/XXXXXXX.XXXXXXX

# 1 Introduction

## 1.1 Background

Data generated by mobile devices, including IoT devices, is projected to reach 79.4 zettabytes (ZB) by 2025, according to a recent report [1]. This vast amount of data is pivotal for enabling next-generation applications such as intelligent transportation systems, smart industries, healthcare, and smart surveillance [2, 3]. Data sharing creates a mutually beneficial scenario, as it enhances the capability of platforms in data markets that operate these emerging applications to deliver more personalized, efficient, and innovative services while providing users with more relevant experiences and tailored product offerings. The ongoing digital transformation is driving the integration of artificial intelligence, machine learning, and other advanced technologies, leading to new business models and significant impacts across various industries [4].

Conventional centralized machine learning approaches face a fundamental challenge of privacy leakage, as they require the transfer of data from end devices to a centralized third-party server for training. Additionally, centralized machine learning may not be viable in cases where data is extremely large and distributed across multiple locations[5]. *Federated learning* addresses these issues by enabling multiple users to collaboratively train a shared global model while keeping their data localized, thereby enhancing privacy and security [6]. This paradigm has been extensively studied in recent years across various domains, including healthcare [7], finance [8], personalized recommendations [9], and smart home applications [10].

On the other hand, as privacy regulations and user rights continue to evolve, there is a growing demand for transparency and clarity in data handling practices. Prominent examples include the European Union's General Data Protection Regulation (GDPR) [11] and the California Consumer Privacy Act (CCPA) [12]. Notably, the GDPR introduces *the right to be forgotten*, which empowers users to request the deletion of their data from platforms, ensuring that personal data is removed not only from storage but also from any further use [13]. While federated learning offers some level of privacy by avoiding the sharing of raw data, it falls short of meeting such stringent requirements. *Machine unlearning* [14, 15] is an emerging technique designed to selectively and efficiently remove the influence of specific data points or users from a trained model without necessitating complete retraining. It essentially reverses previous machine learning processes by modifying or erasing learned models or data. This approach addresses key privacy, security, and compliance concerns, enabling machine learning systems to be updated while maintaining data integrity.

A questionnaire survey of 150 participants in this work demonstrates their satisfaction with the implementation of unlearning. Despite recent regulations and promising technological advancements in machine unlearning, there remains a significant gap in understanding its economic implications. Key questions remain unanswered, such as whether offering an unlearning option can attract

more users and foster greater data sharing, or whether the implementation of unlearning increases operational costs (e.g., through the removal of data) or leads to cost savings by enhancing user incentives for platforms in data markets. Additionally, the broader effects on business models, profitability, and competitive advantage are still underexplored. Without a comprehensive analysis of how machine unlearning impacts the platforms that implement it, its practical deployment remains a challenge.

## 1.2 Challenges

Although federated learning and model sharing create a mutually beneficial scenario in digital economies, users' ability to make informed decisions about their privacy is often severely limited due to their privacy concerns. On one hand, users tend to be risk-averse [16]. After participating in local training and model sharing, they may receive immediate, tangible benefits such as discounts or enhanced personalized services. However, they frequently face imperfect or asymmetric information about when their data is collected, for what purposes, and with what potential consequences [17]. On the other hand, users are highly sensitive to privacy leakage, often preferring that platforms possess minimal information about them [18, 19]. Consequently, users may hesitate to engage in federated learning due to the uncertainty surrounding potential benefits and the risks of privacy loss.

Contract design is an essential economic method to incentivize user model sharing and elicit necessary unlearning processes. A contract ensures that users' training efforts contribute positively to the desired outcomes [20–22]. In our case, it specifies the users' potential benefits, such as monetary compensation, improved services, or access to a personalized machine learning model, thereby ensuring that users understand what they will gain from participating in model sharing. From a legal and ethical standpoint, the contract serves as a formal agreement that helps ensure platform compliance with privacy regulations such as GDPR and CCPA, while also fostering trust between the platform and its users.

When unlearning is available to users, granting privacy-sensitive users the right to be forgotten, the business model and underlying contract design must adapt. Intuitively, unlearning can be seen as a special form of *insurance* for users participating in federated learning. By requesting unlearning, users can retract their shared data and exit the federated learning platform, thereby preventing significant harm from privacy leakage incurred during the learning process. This flexibility may further incentivize greater user participation in federated learning. Survey results in Figure 2 support this hypothesis, showing that participants in both online services are more willing to share their data when they know it can be revoked. However, how unlearning influences the decision-making process of users remains an open question, leading to our first key question:

QUESTION 1. *How will federated unlearning influence privacy-sensitive user's willingness to participate in federated learning?*

If users request unlearning, unlike those who directly leave the platform, the platform must bear the associated unlearning costs[23, 24]. This requires a proper incentive structure guiding users' unlearning decisions, excessive unlearning requests could impose significant burdens on the platform. Significant challenges

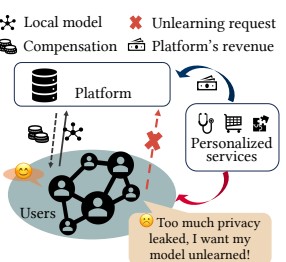
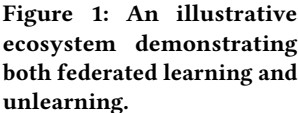

**Figure 1: An illustrative ecosystem demonstrating both federated learning and unlearning.**

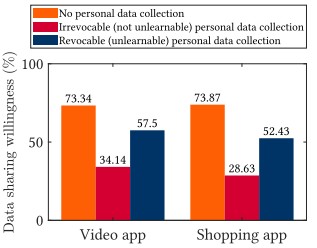

**Figure 2: A survey on users' willingness to share data in two online service applications, with three distinct personal information usage settings.**

arise in designing contracts that incorporate unlearning. The platform must jointly design compensation schemes for both learning and unlearning, striking a balance between encouraging user participation in training and managing the potential costs linked to unlearning. On the other hand, users' local training processes are unobservable or costly to verify [25, 26], and, therefore, cannot be directly contracted. As a result, the contract serves as an indirect mechanism to incentivize users' sequential decisions. This brings us to our second key question:

QUESTION 2. *How should one design an optimal contract that jointly incentivizes user participation in learning and properly elicits unlearning decisions?*

Federated learning platforms can derive several economic benefits from the global model generated through federated learning [27]. For instance, these models' enhanced accuracy and generalization capabilities offer deeper insights into individual user behavior and preferences, which can be leveraged to inform marketing strategies and improve customer service. Furthermore, platforms can monetize these highly accurate and robust models by offering them as premium services or products. The models can be licensed or sold to other businesses, generating additional revenue streams. However, users' risk aversion to uncertainty and sensitivity to privacy loss have a direct influence on the platform's revenue and incentive costs. When unlearning is introduced, the equilibrium between the platform's objectives and the users' participation shifts, as the platform must account for unlearning being available to users, which influences the platform's overall profitability. This raises our third key question:

QUESTION 3. *How do users' risk aversion and privacy sensitivity impact the platform's profitability when unlearning is an available option?*

## 1.3 Main Work and Contributions

We summarize our key novelty and contributions below.

- **Questionnaire Surveys.** We conducted a questionnaire survey to quantify the varying levels of privacy sensitivity among participants, confirm their tendency toward risk

aversion, and demonstrate the incentive effect of unlearning. Due to space limitations, the detailed hypotheses, questionnaire, and results are presented in Appendix A.

- **Modeling and problem formulation.** We propose a comprehensive model for privacy risk that accounts for the platform's data utilization and users' training process. Additionally, we model the users' privacy preferences through their risk aversion and privacy sensitivity. We formulate contract design problems for both unlearning-disabled and unlearning-enabled scenarios. To the best of our knowledge, this is the first framework for studying the incentive effect of unlearning and strategic interactions in federated learning while incorporating unlearning.
- **Optimal contract design and analysis.** We characterize the optimal contract structure, enabling the point-wise decomposition of the optimization problem for each value of privacy leakage, making it tractable to solve. We then analyze the optimized contract design and evaluate the platform's profitability when unlearning is incorporated. We further show that i) the incentive effects of unlearning grow quadratically with users' privacy sensitivity, and ii) enabling unlearning may even profit more than disabling, under higher cost elasticity of risk distribution.
- **Numerical results.** We conduct numerical studies to explore the impacts of various factors, including users' risk aversion, privacy sensitivity, and the platform's revenue and unlearning cost models. We show that the platform's profitability is primarily influenced by users' privacy sensitivity; when users are relatively highly privacy-sensitive, enabling unlearning can significantly improve profitability.

## 2 Related Work

There is a substantial body of literature on federated learning, encompassing both algorithm design [28, 29] and mechanism design [30, 31]. In this section, we primarily focus on reviewing the literature related to machine unlearning.

Machine unlearning was first introduced in [14]. Since then, a growing body of research has focused on developing unlearning techniques. These approaches can be broadly categorized as model-agnostic (e.g., differential privacy [32], knowledge adaptation [33]), and model-intrinsic (e.g., linear regression models [34], Random Forest models [35]). In recent years, *federated unlearning* [36, 37] has emerged as a response to the challenges of data erasure in the federated learning context, where the decentralized nature of data poses new complexities.

Only a limited number of studies explored the economics of federated unlearning. The most relevant work to ours includes [38], which employs a four-stage non-cooperative game to model the interactions and information dynamics between the platform and users during both the learning and unlearning processes. Additionally, [39] investigates users' strategic behavior in permitting partial data revocation in federated unlearning, focusing on the trade-offs between model performance, data privacy, and unlearning costs.

These studies differ from ours in several key aspects. First, the privacy risk models in [38] and [39] assume a deterministic relationship, where users can determine their privacy loss before local training. In contrast, we consider a more realistic scenario where privacy leakage is uncertain before users participate in federated learning. Furthermore, in [38], the asymmetric information concerns users' data profiles, with the platform's goal being to design a mechanism that reveals users' privacy and training costs. Thus, the impact of unlearning on the behavior of risk-averse users and its influence on platform profitability remain largely unexplored, which is the main focus of this work.

## 3 Model and Preliminary

In this section, we first introduce the uncertain privacy leakage model in federated learning. We then provide a detailed discussion of the principal-agent model in federated learning, followed by the formulation of contract design problems. We summarize the key notations in Table 1 in Appendix B.

### 3.1 Federated Learning Setting

*3.1.1 Federated learning overview.* We consider a set of $I$ individual users represented as $\mathcal{I} = \{1, \cdots, I\}$, and a federated learning platform, denoted by $P$, which collects the local training model shared by users and aggregates them to enhance the performance of the global model. Each user $i \in \mathcal{I}$ processes private information $X_i \in \mathcal{X}_i$ that they seek to protect. This private information may include personal characteristics in social networks, transaction details in financial markets, and geolocation data.

Each user $i$'s local data is a noisy realization of the private information $X_i$, and we denote the instance space as $\mathcal{D}_i$ for user $i$. The local training dataset $\boldsymbol{D}_i$ consists of $n_i$ independent and identically distributed (i.i.d.) samples $D_i^\kappa \in \mathcal{D}_i, \kappa \in \{1, \cdots, n_i\}$.

Let $\mathcal{W}$ represent the hypothesis space, and $\ell : \mathcal{W} \times \mathcal{D} \to \mathbb{R}^+$ be a nonnegative loss function. The objective of each user in federated learning is to minimize the empirical risk through local training by solving the following optimization problem:

$$\min_{W_i \in \mathcal{W}} \frac{1}{n_i} \sum_{\kappa=1}^{n_i} \ell(D_i^\kappa, W_i), \tag{1}$$

where $D_i^\kappa$ denotes the $\kappa$-th data sample of user $i$.

We consider a warm-start scenario [40], where the platform already had a pre-trained global model $W_P$, derived from a dataset $D_P$ consisting of $n_P$ data points sampled from $\mathcal{D}_P$. At each iteration round for the global model, after each user $i \in \mathcal{I}$ completes the local training, the platform collects each trained model $W_i$ for aggregation to produce a new global model.

*3.1.2 Local training effort.* After participating in federated learning, each user $i \in \mathcal{I}$ chooses a training effort level $e \in \mathcal{E} = [0, 1]$ for local training as described in (1). This effort includes the computational resources, time, and energy they invest, and we denote the associated costs as $c(e)$. We define the user's local training process as $\Phi : \mathcal{E} \times \mathcal{D} \to \mathcal{W}$. When a user exerts no effort ($e = 0$), it indicates no local training is performed. Conversely, as users increase their local training effort, the resulting model $W_i$ captures more detailed information about the underlying private data $X_i$.

The training effort is often unobservable or costly to verify because local training takes place independently on users' devices, which vary in computational capabilities and energy constraints due to device heterogeneity.

*3.1.3 Privacy risk.* Although federated learning ensures that the raw private dataset $D_i$ does not leave the training devices of the user $i$, several potential privacy risks remain inherent in federated learning [41]. Additionally, users in the social network exhibit correlations with each other[42], and attackers can use this information to amplify the privacy leakage.

We quantify the user's privacy leakage using conditional mutual information[43], as defined in Definition 1.

DEFINITION 1. *For a user $i \in \mathcal{I}$ with private information $X_i \in \mathcal{X}_i$, holding the local training dataset $D_i$, and participating in federated learning via local training with $\Phi(e, D_i)$, the verifiable leaked privacy is defined as:*

$$R(e, D_i) = \delta \cdot \mathrm{MI}\left(X_i; \Phi(e, D_i) \mid W_P\right), \quad (2)$$

*where $\delta \in [0, 1]$ and mutual information $\mathrm{MI}(\cdot; \cdot \mid \cdot)$ measures how much information local training reveals about $X_i$, given that the platform already has access to $W_P$.*

The platform and users may evaluate privacy leakage through techniques such as membership inference attacks [44] and backdoor attacks [45]. We introduce a ratio $\delta \in [0, 1]$ in Definition 1 to further characterize the potentially limited attack capability and the imperfect information about the underlying distribution $\mathcal{X}_i$.

The privacy risk arises before the user participates in federated learning and performs local training via $\Phi(e, D_i)$. Neither the platform nor the user has complete knowledge of each other's data profiles or data correlation. This incomplete information introduces randomness and hence *risk* regarding privacy leakage for both parties. While the exact value of privacy leakage remains uncertain before users engage in local training, both the platform and the users are aware of the probability distribution. We denote $f(R|e)$ as the privacy leakage density function conditional on the user's training effort $e$.

*3.1.4 Federated unlearning.* When unlearning is enabled, the user can request that the platform unlearn their data after local training and model sharing. The goal of unlearning is to remove the influence of a particular user's data from the global model without requiring a complete retraining of the model. We assume an exact unlearning algorithm [24, 46], which ensures that after unlearning, the platform no longer retains any information about the shared model produced during local training $\Phi(e, \mathbf{D}_i)$. Consequently, the privacy leakage defined in Definition 1 is reduced to 0.

During the unlearning process, when a user submits an unlearning request, the platform incurs costs denoted by $q(R)$. These costs include the degradation in the global model's performance and the additional training required to restore the model after unlearning.

EXAMPLE 1. *An example distribution of $X_i$ is a normal distribution, i.e., $X_i \sim \mathcal{N}(x_i, \sigma_i^2)$. This personalized data is modeled as $D_i = X_i + Z_i$, where $Z_i$ is an independent random variable also following a normal distribution $\mathcal{N}(0, \tau_i^2)$. A commonly used model for analyzing the theoretical performance of machine learning systems [47] defines the loss function as $\ell(D, W) = \|D - W\|^2$, where $D$ denotes the training data sample, $W$ represents the machine learning model, and $\| \cdot \|$ is the Euclidean norm. Consider the correlation coefficient of $(X_i, X_P)$*

as $\rho_i$, the privacy leakage is then given by

$$R(e, D_i) = \frac{\delta}{2} \log(1 + \tilde{y} \cdot e), \quad \forall e \in \mathcal{E}, \quad (3)$$

*where*

$$\tilde{y} = \frac{\sigma_i^2 n_i \left[(1 - \rho_i^2) n_P \sigma_P^2 + \tau_P^2\right]}{\tau_i^2 \left(n_P \sigma_P^2 + \tau_P^2\right)} \geq 0, \quad and \ R \in \mathcal{R}. \quad (4)$$

Figure 10 in Appendix B.1 illustrates the distribution of uncertain privacy leakage $R$ conditional on effort $e$.

In the following analysis, we assume that externalities do not exist, meaning one user's decisions do not influence the decisions of others. This assumption is valid because techniques such as secure aggregation and averaging ensure that each user's contribution to the global model and platform is negligible [48, 49]. We will therefore drop the index $i$ in the following.

## 3.2 Principal-Agent Model

In federated learning, the platform serves as the principal, delegating the task of training a machine learning model to the user in each iteration round without directly accessing its data. The user, acting as an agent, retains autonomy over its data and contributes to the learning process by training the local model based on its local data. They also have their own interests, such as preserving privacy or earning incentives for participation.

We examine two scenarios for comparison based on the implementation of machine unlearning: *the unlearning-enabled scenario* and *the unlearning-disabled scenario*. We use the subscript $(\cdot)_D$ and $(\cdot)_E$ to denote variables associated with the unlearning-disabled case and the unlearning-enabled case, respectively.

*3.2.1 Contract variables.* As we discussed in Section 3.1, the user incurs local training costs and faces risks of privacy loss. Without sufficient compensation, it may lack motivation to contribute to the learning process. Compensation serves as an incentive for participation and can take various forms, such as monetary rewards or non-monetary benefits. For instance, the platform may offer efficiency bonuses or enhanced personalized services to the user [17].

For the unlearning-disabled scenario, the contract is defined as $C_D = \{t(R)\}$, where the positive contract term $t(R)$ represents the compensation contingent on the observed privacy leakage $R$ after user's local training and model sharing. In the unlearning-enabled scenario, the contract is denoted as $C_E = \{t(R), \hat{t}(R)\}$, where the additional term $\hat{t}(R) \geq 0$ accounts for compensation to the user when requesting unlearning. If $\hat{t}(R) \equiv 0$, it implies that the user receives no compensation for unlearning.

*3.2.2 Interaction framework.* We model the interactions between the platform and the user as a three-stage game, as in Figure 3.

Specifically, in Stage 0, the platform designs and announces contract $C_D$ when disabling unlearning or $C_E$ when enabling unlearning. The interaction then progresses to Stage 1, where the user determines its level of training effort $e$, which incurs a corresponding cost $c(e)$. This effort impacts the potential privacy leakage associated with the shared data. In Stage 2, after observing the realized privacy leakage $R$ and considering the compensation offered,

| Stage 0: Contract Stage |
| --- |
| The platform specifies either contract $\mathcal{C}_D$ or $\mathcal{C}_E$. If $\mathcal{C}_E$ is chosen, unlearning is enabled, and the user has the option to request unlearning in Stage 2. |

| Stage 1: Learning Stage |
| --- |
| The user selects its training effort $e \in \mathcal{E}$. |

| Stage 2: Unlearning Stage |
| --- |
| The uncertain privacy leakage $R$ is realized. If contract $\mathcal{C}_E$ was specified in Stage 0, the user makes an unlearning decision $a \in \{0,1\}$. |

**Figure 3: The three-stage interaction among the platform and users under two scenarios based on the implementation of unlearning.**

the user on an unlearning-enabled platform decides whether to request unlearning. This decision is represented by the binary choice variable $a(R) \in \{0, 1\}$, where $a(R) = 1$ signifies a request for unlearning. Finally, the platform compensates the user based on the realized privacy leakage $R$ and the pre-agreed terms of the contract. If unlearning is enabled, the compensation also accounts for the user's unlearning decision. Through these stages, the platform and the user complete the execution of the contract.

*3.2.3 Risk-averse user.* The user values its privacy sensitivity as $p \geq 0$, which reflects their attitude towards privacy leakage [50]. For instance, a larger $p$ implies a stronger preference to keep private data confidential. Let $u(\cdot)$ represent the user's utility function. We assume that utility $u(\cdot)$ and training costs $c(e)$ (also referred to as disutility) are additively separable, which is a common assumption in contract theory involving moral hazard [51].

The user's net utility $U_D$ in the unlearning-disabled platform conditional on realized $R$ is given by

$$U_D(R, \mathcal{C}_D, e) = u(t(R) - pR) - c(e). \tag{5}$$

In the unlearning-enabled platform, the user can request unlearning, which eliminates the privacy loss (i.e., $R = 0$), while receiving compensation $\hat{t}(R)$ as specified in the contract $\mathcal{C}_E$. Given the sequential actions of exerting effort $e$ and making the unlearning decision $a(R)$, the user's net utility $U_E$ in the unlearning-enabled platform is expressed as:

$$U_E(R, \mathcal{C}_E, e, a(R)) = u\left(a(R) \cdot \hat{t}(R) + (1 - a(R)) \cdot (t(R) - pR)\right) - c(e). \tag{6}$$

Due to their limited computational resources, the user is assumed to be risk-averse, as described by the Arrow–Pratt measure of risk aversion [52, 53], formalized in Definition 2.

DEFINITION 2. *A user is risk averse if*

$$\mathbb{E}[u(\tilde{\pi})] \geq u(\mathbb{E}(\tilde{\pi})), \tag{7}$$

*for every risky payoff $\tilde{\pi}$. The Arrow-Pratt measure of absolute risk aversion is given by*

$$r(\pi) \triangleq -\frac{u''(\pi)}{u'(\pi)}. \tag{8}$$

Moreover, as stated in [52, 53], risk aversion corresponds to the concavity of the utility function. Thus, the utility function $u(\cdot)$ is strictly concave, with $u' > 0$ and $u'' < 0$. We also assume the cost function $c(\cdot)$ is strictly convex, with $c' > 0$ and $c'' > 0$.

*3.2.4 Risk-neutral platform.* The platform's revenue generated by exploiting users' privacy-sensitive data is denoted as $S(R)$, which is an increasing function in $R$. This model generalizes the one in [17] and captures that the platform can benefit from privacy leakage in several significant ways, such as price discrimination, target advertising, and interest rate setting in credit markets.

Additionally, the platform typically is a big organization, thus the uncertainty from any single user is diluted, justifying the assumption of a risk-neutral platform. We define the surplus of the unlearning-disable platform as $V_D$, which is the difference between the platform's revenue $S(R)$, and compensation cost according to the agreed-uponed contract. Therefore, when unlearning is disabled, the platform's surplus is expressed as:

$$V_D(R, \mathcal{C}_D) = S(R) - t(R). \tag{9}$$

When unlearning is enabled, if the user requests unlearning, the revenue from that user's privacy leakage drops to zero, and the platform is required to pay $\hat{t}(R)$ to the user and bear the unlearning cost $q(R)$. Given the user's unlearning decision $a(R)$, the surplus of the unlearning-enabled platform $V_E$, is given by:

$$V_E(R, \mathcal{C}_E, a(R)) = a(R) \cdot \left(-q(R) - \hat{t}(R)\right) + (1 - a(R)) \cdot (S(R) - t(R)). \tag{10}$$

## 3.3 Contract Design Problem

To incentivize user's effort and elicit the proper unlearning decision, we consider contract design for the platform. Since the effort is the user's hidden action, not directly observable by the platform, it leads to a moral hazard problem [54].

*3.3.1 The unlearning-disabled scenario.* In this scenario, the platform's objective is to design an optimal contract, denoted by $\mathcal{C}_D$, that incentivizes the user to exert the desired effort level $e$. This contract aims to maximize the platform's expected surplus, represented as $\mathbb{E}_R[V_D(R, \mathcal{C}_D) \mid e]$, which takes into account the user's expected utility based on their training effort, denoted as $\mathbb{E}_R[U_D(R, \mathcal{C}_D, e) \mid e]$. The platform's optimization problem can be formulated as follows:

$$\max_{e, \mathcal{C}_D} \quad \mathbb{E}_R[V_D(R, \mathcal{C}_D) | e] \tag{11a}$$

$$\text{s.t.} \quad \mathbb{E}_R[U_D(R, \mathcal{C}_D, e) \mid e] \geq 0, \tag{11b}$$

$$e \in \arg\max_{\bar{e} \in \mathcal{E}} \mathbb{E}_R[U_D(R, \mathcal{C}_D, \bar{e}) \mid \bar{e}], \tag{11c}$$

$$t(R) \geq 0, \forall R \in \mathcal{R}. \tag{11d}$$

The constraint in (11b) represents the individual rationality (IR) condition. It ensures that the user's expected net utility from participating in the federated learning platform and exerting effort cannot fall below its reservation utility, which is its net utility when exerting no effort (i.e., $e = 0$). The constraint in (11c) is the incentive compatibility (IC) condition. This ensures that even though the platform cannot directly observe the user's effort, the user is incentivized to choose the effort level that aligns with the platform's objectives. Finally, the constraint in (11d) enforces limited liability [55], ensuring that the user's compensation remains non-negative for any realized $R \in \mathcal{R}$.

### 3.3.2 The unlearning-enabled scenario.

When unlearning is enabled, the platform must design a contract that not only incentivizes the user's training effort $e$ but also elicits the user's sequential decision-making regarding unlearning, represented by $a(R)$.

In Stage 2, the privacy leakage $R$ is realized, the user chooses $a(R)$ to maximize its utility by deciding whether to unlearn:

$$a(R) \in \underset{a \in \{0,1\}}{\arg \max} \; a \cdot \hat{t}(R) + (1-a) \cdot (t(R) - pR), \quad \forall R \in \mathcal{R}. \quad (12)$$

By backward induction, anticipating the future realization of privacy leakage $R$ alongside (12), the expected inter-temporal utility for the user is given by $\mathbb{E}_R [U_E(R, C_E, e) \mid e, a(R)]$. The platform's expected surplus, based on the user's sequential decision $(e, a(R))$, is given by $\mathbb{E}_R [V_E(R, C_E) \mid e, a(R)]$. Then the contract design problem in the unlearning-enabled scenario is as follows:

$$\max_{e, C_E, a(R)} \quad \mathbb{E}_R [V_E(R, C_E) \mid e, a(R)] \quad (13a)$$

$$\text{s.t.} \quad \mathbb{E}_R [U_E(R, C_E, e) \mid e, a(R)] \geq 0, \quad (13b)$$

$$e \in \underset{\bar{e} \in \mathcal{E}}{\arg \max} \; \mathbb{E}_R [U_E(R, C_E, \bar{e}) \mid \bar{e}, a(R)], \quad (13c)$$

$$a(R) \in \underset{a \in \{0,1\}}{\arg \max} \; a \cdot \hat{t}(R) + (1-a) \cdot (t(R) - pR), \; \forall R \in \mathcal{R}, \quad (13d)$$

$$t(R) \geq 0, \; \hat{t}(R) \geq 0, \; \forall R \in \mathcal{R}. \quad (13e)$$

Compared to the unlearning-disabled contract design in (11), the unlearning-enabled contract design introduces additional complexity, as it requires incentive compatibility across two stages, resulting in the user's expected utility being inter-temporal.

### 3.3.3 Challenges.

The contract design problem in the unlearning-disabled scenario falls into the standard moral hazard problem with bounded payments [55]. However, directly comparing it to the unlearning-enabled scenario is inherently complex due to the additional decision-making process introduced by unlearning. First, both the unlearning-enabled platform and the user must consider the initial effort $e$ and subsequent unlearning decision $a(R)$, which are interdependent. The compensations for learning $t(R)$ and unlearning $\hat{t}(R)$ are also coupled. The contract $C_E$ must be designed to ensure that the user's optimal decisions—regarding both effort and unlearning—align with the platform's goal of surplus maximization.

## 4 Optimal Contract Design and Analysis

In this section, we first analyze the incentive effects of unlearning in survey results. Then we examine the contract design problem.

### 4.1 The Incentive Effect of Unlearning

Survey results in Appendix A indicate that the introduction of unlearning can significantly increase users' willingness to participate in federated learning and model sharing. To further quantify these findings, we consider a simplified scenario in which the user's training effort is binary, i.e., $\mathcal{E} = \{0, \bar{e}\}$. The platform offers a constant compensation for the user's effort $\bar{e}$, denoted as either $\bar{t}_D$ for the unlearning-disabled setting and $\bar{t}_E$ for the unlearning-enabled setting.

In the unlearning-disabled platform, the platform's problem in (11) is reduced to determine the minimum compensation $\bar{t}_D$ that satisfies the user's individual rationality (IR) constraint. The minimum compensation $\bar{t}_D$ is given by:

$$\bar{t}_D = \inf \left\{ \bar{t} \geq 0 : \int_{\mathcal{R}} u(\bar{t} - pR) \cdot f(R \mid \bar{e}) dR - \bar{c} \geq 0 \right\}, \quad (14)$$

where $pR$ represents the privacy cost associated with the realized leakage $R$, and $\bar{c}$ denotes the effort cost.

In the unlearning-enabled platform, if the realized privacy leakage $R$ leads to $\bar{t}_E - pR < 0$, the user will request unlearning to ensure their utility remains non-negative. Consequently, the platform must adjust its compensation to account for the possibility of unlearning. Thus, the problem in (13) is reduced to obtaining the minimum compensation $\bar{t}_E$:

$$\bar{t}_E = \inf \left\{ \bar{t} \geq 0 : \int_{\mathcal{R}} u(\max\{\bar{t} - pR, 0\}) \cdot f(R \mid \bar{e}) dR - \bar{c} \geq 0 \right\}. \quad (15)$$

We further define the incentive effect in Definition 3.

**Definition 3.** *The incentive effect (IE) is the relative difference in compensation required between the unlearning-disabled and unlearning-enabled settings, expressed as:*

$$\text{IE} = \frac{\bar{t}_D - \bar{t}_E}{\bar{t}_E}. \quad (16)$$

The ratio $\frac{\bar{t}_D - \bar{t}_E}{\bar{t}_E}$ in Definition 3 captures how the introduction of unlearning creates an additional incentive for user participation. We are ready to present the following theorem:

**Theorem 1.** *The optimal compensation required to incentivize user participation in the unlearning-disabled and unlearning-enabled settings is bounded by:*

$$\bar{t}_D - \bar{t}_E \geq 0 \quad and \quad \frac{\bar{t}_D - \bar{t}_E}{\bar{t}_E} = \Omega(rp^2). \quad (17)$$

The proof of Theorem 1 is presented in Appendix D.1. This theorem suggests that the unlearning-disabled platform must always offer a higher compensation ($\bar{t}_D$) to induce the same level of effort from the user as compared to the unlearning-enabled platform ($\bar{t}_E$). This implies that unlearning introduces additional incentive for learning. Specifically, in the unlearning-enabled setting, the user's expected utility increases since its potential losses from privacy leakage are capped due to the option to unlearn.

Additionally, the lower bound of the incentive effect is given by $\Omega(rp^2)$, indicating that the incentive effect grows quadratically with the user's privacy sensitivity $p$ and linearly with its risk aversion $r$. In other words, as the user become more risk-averse and/or more privacy-sensitive, the unlearning-enabled platform requires relatively lower compensation to incentivize its participation.

### 4.2 Optimal Contract Design

#### 4.2.1 The general case.

In this subsection, we aim to solve the general contract design problem when user's effort is unobservable by the platform. As mentioned, the main challenge is the joint design of the compensation $t(R)$ for learning and the compensation $\hat{t}(R)$ to incentivize the user's sequential actions in alignment with the platform's overall objectives.

To address this complexity, we present Theorem 2 to characterize the optimal contract structure.

**Theorem 2.** *For any realized privacy leakage $R \in \mathcal{R}$, the optimal contract structure is as follows:*

- *If $R \in \mathcal{R}_U = \left\{ R \mid p \geq \frac{S(R)+q(R)}{R} \right\}$, the platform should design the contract such that $S(R) + q(R) \leq t(R) - \hat{t}(R) \leq pR$ to incentivize the user to request unlearning.*

- *If $R \in \mathcal{R}_L = \left\{ R \mid p < \frac{S(R)+q(R)}{R} \right\}$, the platform should structure the contract so that $pR \leq t(R) - \hat{t}(R) \leq S(R) + q(R)$ to encourage the user to retain their shared model without requesting unlearning.*

This contract structure ensures that the user's decision to unlearn or not is aligned with the platform's objective of maximizing the expected surplus. By managing the trade-offs between learning and unlearning costs, the platform can optimize its overall surplus. The proof of Theorem 2 is provided in Appendix D.2, and the optimal contract structures for each $R$ are illustrated in Figure 14.

The theorem demonstrates that the platform's optimal incentive strategy for each level of privacy leakage $R \in \mathcal{R}$ depends on the user's privacy sensitivity and the platform's combined marginal revenue and unlearning costs. Based on Theorem 2, the joint design problem in (13) can be separated into two distinct cases: one for $R \in \mathcal{R}_U$ and the other for $R \in \mathcal{R}_L$.

Furthermore, Theorem 2 characterizes the optimal contract structure for the joint design of $t(R)$ and $\hat{t}(R)$ at each level of privacy leakage. Specifically, for $R \in \mathcal{R}_U$, the compensation $\hat{t}(R)$ is applied, while $t(R)$ is not uniquely defined as long as the difference $t(R) - \hat{t}(R)$ falls within the interval $[S(R)+q(R), pR]$. This flexibility allows the contract designer to optimize $\hat{t}(R)$ in the $R \in \mathcal{R}_U$ case, and vice versa for $t(R)$.

*4.2.2 The special case: Binary privacy leakage.* We then examine a binary privacy leakage setting based on Theorem 2, which allows us to derive the closed-form optimal contract for more insights.

In the binary privacy leakage setting, there are only two possible privacy risk outcomes, $\{R_H, R_L\}$ with $R_H > R_L \geq 0$, representing high and low leakage, respectively. The user's effort influences the probabilities of these outcomes, with $F(e) \triangleq \Pr(R_H|e)$. To capture the relationship between training cost and privacy leakage, we define the *cost elasticity of risk distribution* in Definition 4.

**Definition 4.** *Define the cost elasticity of risk distribution ($E_{CR}$) as the elasticity ratio of the training cost to the probability of high privacy leakage with respect to the user's local training effort $e$ as:*

$$E_{CR}(e) = \frac{c'(e)/c(e)}{F'(e)/F(e)}. \tag{18}$$

Definition 4 measures how responsive the training cost is relative to the probability of high privacy leakage as the user adjusts $e$.

In this case, the platform's revenues become $\{S_H, S_L\}$, and the unlearning costs become $\{q_H, q_L\}$. Considering the platform's strictly convex revenue function and unlearning cost function, we have the condition $\frac{S_L+q_L}{R_L} < \frac{S_H+q_H}{R_H}$. For the convex unlearning cost model, this may be due to the complexity of removing deeply embedded or widely distributed data.

Further details of this setting are provided in Appendix D.3. Let the expected surpluses of the platform be denoted as $\mathbb{E}[\bar{V}_D]$ for the unlearning-disabled platform and $\mathbb{E}[\bar{V}_E]$ for the unlearning-enabled platform. We now present Theorem 3.

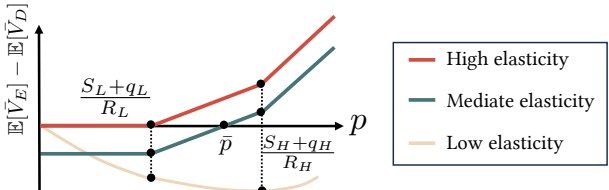

**Figure 4: The expected surplus difference $\left( \mathbb{E}[\bar{V}_D] - \mathbb{E}[\bar{V}_E] \right)$ under effort $e$ versus user's privacy sensitivity $p$.**

**Theorem 3.** *To incentivize the user to exert a fixed effort $e \in \mathcal{E}$:*

- *If the incentivized effort $e$ satisfies $E_{CR}(e) < 1 - \frac{u(-pR_L)}{c(e)}$ (high elasticity), the surplus difference $\left( \mathbb{E}[\bar{V}_D] - \mathbb{E}[\bar{V}_E] \right)$ increases piecewise linearly in $p$.*

- *If the incentivized effort $e$ satisfies $E_{CR}(e) \geq 1 - \frac{u(-pR_L)}{c(e)}$ (low elasticity), the surplus difference $\left( \mathbb{E}[\bar{V}_D] - \mathbb{E}[\bar{V}_E] \right)$ is strictly negative and decrease in $p$ until $p > \frac{S_H+q_H}{R_H}$.*

Figure 4 illustrates Theorem 3. Detailed proof and further discussion are provided in the Appendix D.3.

When $E_{CR}(e) \leq 1$, the probability of high privacy leakage increases more rapidly with effort than the training cost. In this case, the user is incentivized to exert higher effort to trigger high privacy leakage, maximizing its net utility. Here, the incentive problems for both the unlearning-disabled and unlearning-enabled platforms are similar. However, the unlearning-disabled platform must absorb all potential loss. As the user's privacy sensitivity increases (i.e., $p > \frac{S_L+q_L}{R_L}$), the unlearning-enabled platform can cap its maximum loss at unlearning and compensation for unlearning, while the unlearning-disabled platform's loss decreases linearly with $p$.

When $1 < E_{CR}(e) < 1 - \frac{u(-pR_L)}{c(e)}$, the surplus difference behaves differently. In this scenario, the user exert less effort because the training cost grows more sharply than the probability of high privacy leakage. For small $p \leq \bar{p}$ in Figure 4, the higher compensation for the user's training cost dominates the platform's surplus, resulting in a lower surplus compared to the unlearning-disabled setting. This effect becomes more pronounced when $E_{CR}(e) \geq 1 - \frac{u(-pR_L)}{c(e)}$. These findings are summarized in Observation 1.

**Observation 1.** *Unlearning can improve the platform's profitability when the cost elasticity of risk distribution is high, where the user is less concerned about the training costs.*

The platform then chooses the optimal effort to incentivize the user to exert to maximize the expected surplus. Due to the complexity of obtaining a closed-form solution, we numerically examine this in Section 5 for the continuous leakage case.

## 5 Simulations

In this section, we first perform numerical studies of the incentive effect of unlearning. Then we compare the expected surpluses of two unlearning profiles and investigate the platform's probability under different degrees of risk aversion and privacy sensitivity.

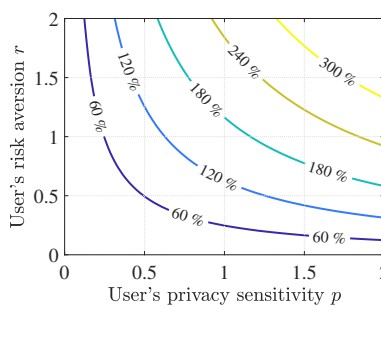
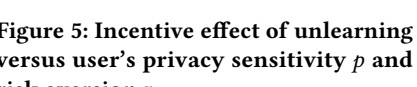

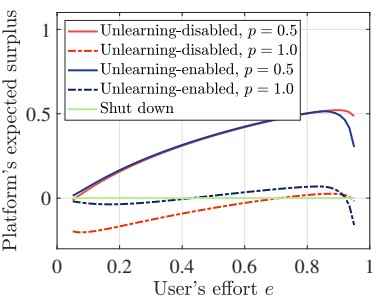

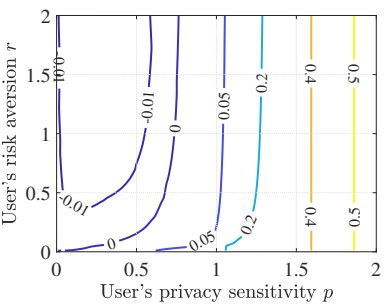

**Figure 5: Incentive effect of unlearning versus user's privacy sensitivity $p$ and risk aversion $r$.**

**Figure 6: Platform's expected surplus versus user's effort when risk aversion $r = 0.5$.**

**Figure 7: The difference of achievable expected surplus when enabling and disabling unlearning.**

## 5.1 Setting

We consider the federated learning setting in Example 1. Assume the random factor $\tilde{y}$ follows an exponential distribution with parameter $\theta = 100$. The utility function is defined as $u(\pi) = 1 - \exp(-r\pi)$, where $r$ represents the user's degree of risk aversion. The training cost is modeled by $c(e) = -\xi \cdot \log(1 - e)$, where $e \in [0, 1)$. For the platform's setting, we consider $S(R) = R^2$ and $q(R) = 0.1 \cdot R^2$. A more detailed setup for simulations is provided in Appendix C.1.

To solve the contract design problems in (11) and (13), we apply the first-order approach[56]. This method simplifies the set of possible incentive constraints to a local incentive constraint, transforming the problem into a standard convex programming one. More details are provided in the Appendix B.2. We then quantify $\mathcal{R}$ to ensure the optimization is solvable using CVX, a package for specifying and solving convex programs [57, 58].

## 5.2 Numerical Results

*5.2.1 Unlearning incentivizes learning under risk aversion.* We begin by investigating the incentive effects of unlearning in relation to two key factors: user risk aversion $r$ and user privacy sensitivity $p$.

Figure 5 illustrates a contour plot demonstrating that as both the user's risk aversion $r$ and privacy sensitivity $p$ increase, the incentive effect also increases. This demonstrates that higher levels of risk aversion and privacy sensitivity lead to a greater disparity in the compensation required to motivate users in the unlearning-disabled versus unlearning-enabled platforms. These results are consistent with the theoretical bound presented in Theorem 1.

*5.2.2 Contract design problem.* We compare the platform's contract design under both unlearning-enabled and unlearning-disabled settings, considering $p \in [0, 2]$ and $r \in [0, 2]$.

Figure 6 shows the platform's expected surplus as the user exerts a fixed effort level $e \in [0.05, 0.95]$ under two privacy sensitivity settings: $p = 0.5$ and $p = 1.0$. When privacy sensitivity is low ($p = 0.5$), the expected surplus of the unlearning-enabled platform is slightly higher than that of the unlearning-disabled one, which aligns with our analysis of the binary privacy leakage case. As the effort increases, reaching the low elasticity region ($e > 0.84$), the expected surplus in the unlearning-disabled setting surpasses that of the unlearning-enabled one.

At higher privacy sensitivity ($p = 1.0$), the surplus difference remains strictly positive until $e > 0.912$, at which point the low elasticity reappears. Additionally, we mark a zero-surplus line, indicating that when privacy sensitivity is high (e.g., $p = 1.0$), the unlearning-disabled platform may frequently ($e \in [0.71, 0.94]$) shut down its federated learning operations due to negative expected surplus. In contrast, the unlearning-enabled platform can sustain a positive expected surplus over a broader effort range $e \in [0.43, 0.922]$.

We further examine the platform's achievable expected surplus concerning user risk aversion and privacy sensitivity in Figure 7. First, an increase in user risk aversion $r$ leads to a slight improvement in expected surplus. This is because when users are more risk-averse, the platform needs to smooth the contract to mitigate the risks users face, thereby lowering the risk premium. In this context, Figure 12 in Appendix C.2 illustrates the optimal compensation schemes for $e = 0.85$. Additionally, unlearning significantly enhances the platform's profitability when user privacy sensitivity $p$ is high. As privacy sensitivity increases, the gap between the achievable surplus in the two settings widens.

We also examine the platform's revenue and incentive cost separately in Appendix C.2.

## 6 Conclusion

To the best of our knowledge, this paper is the first to examine the incentive effects of unlearning on both user behavior and platform profitability. We have proposed a necessary condition that decomposes the intertwined compensation mechanisms for learning and unlearning, thereby making the contract design problem for unlearning-enabled scenarios tractable and solvable. Our formulation and analysis have highlighted several key insights: (i) introducing unlearning can enhance users' willingness to participate in federated learning; (ii) when users' local training processes are unobservable, the incentive effects of unlearning depend on the relationship between training costs and the probability of significant privacy leakage as users exert the incentivized effort; and (iii) higher privacy leakage sensitivity leads to a larger surplus advantage of unlearning compared to disabling unlearning.

In future work, we will explore this direction under more general conditions by considering the externality model, in which case one user's unlearning behaviors may trigger other users to unlearn.

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

# A  Questionnaire Survey

In this section of the Appendix, we present details of the questionnaire survey conducted in October 2024, followed by our findings and insights.

## A.1  Hypotheses

The primary goal is to evaluate the following three hypotheses:

- **H1**: Users of online services have varying levels of privacy sensitivity.
- **H2**: Users tend to exhibit risk aversion when confronted with privacy risk.
- **H3**: Providing an "unlearning" option significantly increases users' willingness to share data.

## A.2  Survey Setup

*A.2.1  Demographics.* To ensure that the data was comprehensive and representative, minimizing biases toward specific groups, we carefully selected five demographic dimensions that are most likely to influence privacy sensitivity or consumer behavior: gender, income, educational background, age, and race, and balance them as much as possible. We applied strict stratified sampling to the first three dimensions to ensure balanced data across these categories.

Specifically, we recruited 150 participants via the Prolific [1]. Our participant pool was relatively balanced by gender with 36% male, 39% female, and 22% identifying as non-binary or third gender. Income levels were also well distributed, with each personal income bracket covering between 13.1% and 24.2% of the sample. In terms of educational background, there was diversity as well, with participants in each category ranging from 17.6% to 38.6%. Regarding age, 56.5% of participants were aged 18-34, and 35.4% were aged 35-55, which are precisely the groups that are frequent internet users. It is important to note that 73.9% of the participants were white, which reflects the broader demographic distribution of Prolific's user base[59] but may introduce a potential bias toward this racial group.

Demographic data for these five dimensions are illustrated in Figure 8.

*A.2.2  Privacy sensitivity metric.* To quantify participants' privacy sensitivity, we used the widely recognized *willingness to pay a premium* (WPP) method, which measures participants' perceived added value of a product (e.g., brand, quality, or features) [60].

To ensure that all "privacy sensitivity" scores fell within the range [0,1], we also apply *Min-Max Normalization* to standardize the values.

*A.2.3  Remark.* Two key points should be noted. First, to ensure participants quickly and clearly understand the concept of "unlearning", we used the phrase "erase your contribution of shared data" instead of more technical terms like "unlearning" or "federated unlearning". Second, to make the survey relevant to real-world applications of federated unlearning, we selected scenarios that reflect typical privacy concerns in practice, such as online shopping platforms and video apps.

## A.3  Survey Questions

*A.3.1  Question Set 1.* Question Set 1 is designed to measure participants' privacy sensitivity and verify our **H1**. Participants were asked how much they would be willing to pay to mitigate the consequences of a privacy breach. We created four scenarios—social media, health fitness app, news website, and ride-sharing app—each involving a potential privacy breach. For each scenario, participants indicated the maximum amount they would be willing to pay to prevent the loss and keep their data private. The specific questions are as follows:

---
[1]Prolific: https://www.prolific.com

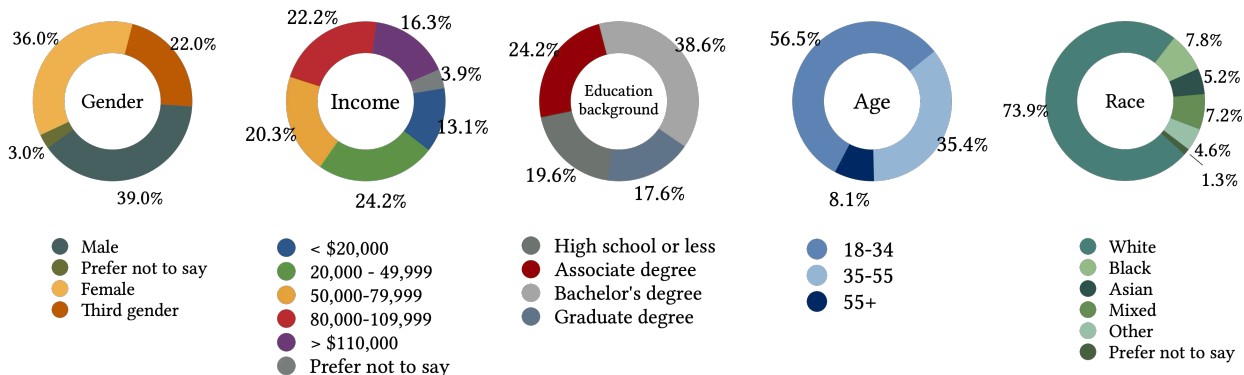

**Figure 8: Demographic distribution across gender, income, educational background, age, and race.**

- You have shared personal data with a social media platform to receive personalized recommendations for accounts or posts. However, you later notice an increase in targeted ads and realize your data may be at risk of privacy breaches. Now, you have the option to delete your data and stop sharing information in the future. What is the maximum amount you would be willing to pay to keep your data private?
- You are using a Health & Fitness App and have provided personal information to receive customized fitness plans. One day, you find out that your data could be sold to third parties or insurers, potentially raising your premiums or increasing targeted ads. What is the maximum amount you would be willing to pay to keep your data private?
- You are using a news website that tracks your reading habits to offer tailored content. However, it may also sell your preferences to third-party advertisers or deliver biased news. What is the maximum amount you would be willing to pay to keep your data private?
- You are using a ride-sharing app that tracks your past locations to suggest faster routes. However, it may also share this data with third parties or adjust prices based on your location and demand (e.g., price surges). What is the maximum amount you would be willing to pay to keep your data private?

*A.3.2 Question Set 2.* Question Set 2 aims to assess participants' risk aversion and explore how the availability of an unlearning option affects their willingness to share data, addressing both **H2** and **H3**. Two specific scenarios—"video app usage" and "shopping app usage"—were presented. Participants were asked to indicate their willingness to share data under the following three conditions:

(1) The app does not actively collect personal information and offers only a basic search function.
(2) The app collects your information, resulting in both positive consequences (such as more personalized video recommendations) and negative consequences (such as more targeted ads and repetitive video suggestions). All consequences, both positive and negative, affect you directly.

(3) Similar to condition (2), but with the added option to request the app to erase your shared information at any time (for example, if you are dissatisfied with the benefits or privacy implications).

The specific questions for these scenarios are as follows:

- There is a video app (such as YouTube) where you can find anything that may interest you (e.g. latest news, music, movie cuts). Please show your willingness of three different choices.
- There is a shopping app (such as Amazon, Walmart) where you can buy various items ranging from living things to snacks. Please show your willingness of three different choices by moving the sliding blocks.

*A.3.3 Attention-check.* To maintain data quality, we also included two attention-check questions. Only participants who passed both checks were included in the final analysis. The attention-check questions for Problem Set 1 and Problem Set 2 are as follows:

- Jack is a wealthy programmer who is solely focused on making money and knows nothing about financial management. With this in mind, select Jack's area of expertise: "Investing in the stock market", "Investing in real estate", "Both", or "Neither".
- Carol is very sensitive in her daily life and does not share any browsing history related to her private life. Which choice would she prefer? "TikTok's intelligent recommendation system based on personal data", "Walmart app's intelligent recommendation system based on browsing records", "Both are acceptable", or "Neither".

## A.4 Survey Results

Out of the 150 participants, 122 successfully passed both attention-check questions, resulting in a pass rate of 81.3%. All subsequent data analyses are based on the responses from participants who passed the attention checks.

*A.4.1 Varying levels of privacy sensitivity.* Participants' privacy sensitivity across the four scenarios, along with the mean values, is

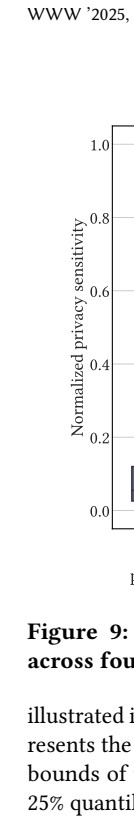

**Figure 9: Normalized privacy sensitivity of participants across four different scenarios.**

illustrated in a box plot in Figure 9. The central line in the figure represents the median value, approximately 0.04. The upper and lower bounds of the box correspond to the 75% quantile (0.10) and the 25% quantile (0.02), indicating that most data points fall within this range. The whiskers extend to the upper limit (0.22) and the lower limit ($1e - 06$), with several outliers shown as orange points. This variation in privacy sensitivity levels among participants supports **H1**.

However, the variation along with the clustering of data around 0.04 may suggest a potential limitation in the survey design. The scenarios might have been too broad, causing participants' responses to be overly influenced by subjective preferences, leading to a wide disparity in selected values.

*A.4.2   Risk Aversion Tendency.* As shown in Figure 2, participants' willingness to share personal data is 73.34% and 73.87% when the application does not actively collect personal information (as indicated by the orange bars in Figure 2). However, when participants were presented with both the positive and negative consequences of data sharing, willingness dropped to 34.14% and 28.63%. These results strongly validate **H2** and support the overarching premise of the study.

*A.4.3   Unlearning Option and Change of Willingness.* Figure 2 also demonstrates the increase in participants' willingness to share data when a revocable data-sharing option was introduced. Willingness to share data increased by 23.26% and 23.8% across the two scenarios. The contrast is visually emphasized by the red and blue bars in Figure 2. This finding suggests that when faced with potential risks, the availability of an unlearning option significantly boosts participants' willingness to share data, directly supporting **H3**.

# B   Analysis Details

In this part of the Appendix, we provide the analysis of Example 1 and the relaxed contract design problem to support our analysis in the main text. We summarize the key notations in Table 1.

## B.1   Analysis of Example 1

The distribution of $X_i$ is assumed to follow a normal distribution, i.e., $X_i \sim \mathcal{N}(x_i, \sigma_i^2)$. This personalized data is modeled as $D_i = X_i + Z_i$, where $Z_i$ is an independent random variable also following a normal distribution, i.e., $Z_i \sim \mathcal{N}(0, \tau_i^2)$. Consequently, the local training data sample $D_i^\kappa \in \mathbf{D}_i$ for user $i$ consists of i.i.d. samples from $\mathcal{N}(x_i, \sigma_i^2 + \tau_i^2)$.

Define the loss function as $\ell(D, W) \triangleq \|D - W\|^2$, where $D$ denotes the training data sample, $W$ denotes the machine learning model, and $\| \cdot \|$ denotes the Euclidean norm. When the user $i$ participates in the federated learning, the well-trained local model based on $\mathbf{D}_i$ is given by:

$$W_i^* = \arg\min_{W \in \mathcal{W}} \frac{1}{n_i} \sum_{\kappa=1}^{n_i} \ell(D_i^\kappa, W) = \frac{1}{n_i} \sum_{\kappa=1}^{n_i} \|D_i^\kappa - W\|^2, \qquad (19)$$

The minimum of this objective function is given by the sample mean. Therefore, the well-trained local model and its underlying distribution can be expressed as:

$$W_i^* = \frac{1}{n_i} \sum_{\kappa=1}^{n_i} D_i^\kappa, \quad W_i^* \sim \mathcal{N}\left(x_i, \sigma_i^2 + \frac{1}{n_i}\tau_i^2\right). \qquad (20)$$

Similarly, the well-trained platform model $W_P^*$ for the warm-start federated learning platform and its underlying distribution can be given by

$$W_P^* = \frac{1}{n_P} \sum_{\kappa=1}^{n_P} D_P^\kappa, \quad W_P^* \sim \mathcal{N}\left(x_P, \sigma_P^2 + \frac{1}{n_P}\tau_P^2\right), \qquad (21)$$

where $D_P^\kappa$ represented the $\kappa$-th data sample to train the global machine learning model $W_P^*$.

Let the correlation coefficient of $(X_i, X_P)$ to be denoted as $\rho_i$. The joint distribution of $\left(X_i, W_i^*, W_P^*\right)$ follows a multivariate normal distribution:

$$\begin{pmatrix} X_i \\ W_i^* \\ W_P^* \end{pmatrix} \sim \mathcal{N}\left(\begin{pmatrix} x_i \\ x_i \\ x_P \end{pmatrix}, \Sigma_i\right), \qquad (22)$$

where

$$\Sigma_i = \begin{bmatrix} \sigma_i^2 & \sigma_i^2 & \rho_i\,\sigma_P\,\sigma_i \\ \sigma_i^2 & \frac{\tau_i^2}{n_i} + \sigma_i^2 & \rho_i\,\sigma_P\,\sigma_i \\ \rho_i\,\sigma_P\,\sigma_i & \rho_i\,\sigma_P\,\sigma_i & \frac{\tau_P^2}{n_P} + \sigma_P^2 \end{bmatrix}. \qquad (23)$$

Then the privacy leakage for the well-trained local and well-trained global models, denoted as the maximum privacy leakage, is

**Table 1: Key notations and descriptions.**

| Symbol | Description |
|---|---|
| $R$ | Privacy leakage modeled by conditional mutual information. |
| $C_D = \{t(R)\}$ | Contract in the unlearning-disabled setting, where $t(R)$ is the compensation for privacy leakage $R$. |
| $C_E = \{t(R), \hat{t}(R)\}$ | Contract in the unlearning-enable setting, where $t(R)$ is the compensation for privacy leakage $R$, and $\hat{t}(R)$ is the compensation for unlearning $R$. |
| $e, c(e)$ | User's local training effort and the corresponding training cost. |
| $a(R)$ | User's unlearning decision when realized $R$ |
| $r$ | User's degree of risk aversion. |
| $p$ | User's privacy sensitivity. |
| $q(R)$ | Platform's unlearning cost for privacy leakage $R$. |
| $S(R)$ | Platform's revenue from privacy leakage $R$. |

given by:

$$(R_i)_{max} = \text{MI}\left(X_i; W_i^* \mid W_P^*\right)$$

$$= H(X_i, W_P^*) + H(W_i^*, W_P^*) - H(X_i, W_i^*, W_P^*) - H(W_P^*)$$

$$= \frac{1}{2} \log \frac{\det\left(\Sigma_{\left(X_i, W_P^*\right)}\right) \det\left(\Sigma_{\left(W_i^*, W_P^*\right)}\right)}{\det\left(\Sigma_{\left(X_i, W_i^*, W_P^*\right)}\right) \det\left(\Sigma_{\left(W_P^*\right)}\right)}$$

$$= \frac{1}{2} \log \left( 1 + \underbrace{\frac{\sigma_i^2 n_i \left[(1 - \rho_i^2) n_P \sigma_P^2 + \tau_P^2\right]}{\tau_i^2 \left(n_P \sigma_P^2 + \tau_P^2\right)}}_{= \tilde{y}} \right).$$

$$(24)$$

The term $\frac{\sigma_i^2 n_i}{\tau_i^2}$ in $\tilde{y}$ corresponds to the user's effort, representing the leaked privacy resulting from the sharing of its local model. The user can increase the number of local data samples (increasing $n_i$) or reduce the noise in the local data (decreasing $\tau_i$) to contribute more privacy leakage, i.e., exert higher training effort.

Finally, integrating the ratio $\delta \in [0, 1]$, the privacy leakage is given by

$$R(e, D_i) = \frac{\delta}{2} \log(1 + \tilde{y} \cdot e), \ \forall e \in \mathcal{E}, \tag{25}$$

where

$$\tilde{y} = \frac{\sigma_i^2 n_i \left[(1 - \rho_i^2) n_P \sigma_P^2 + \tau_P^2\right]}{\tau_i^2 \left(n_P \sigma_P^2 + \tau_P^2\right)} \geq 0, \quad \text{and } R \in \mathcal{R}. \tag{26}$$

## B.2 Relaxed Contract Design Problem

The complexity of solving the moral hazard problem arises from the user's incentive compatibility constraint. The most common method to address this is relaxing it according to the first-order condition. Following [56], we conclude the first-order approach in Lemma 1.

LEMMA 1. *If the platform is risk neutral and the user is risk averse, $U(\pi, e) = u(\pi) - c(e)$ with $u' > 0, u'' < 0, c' > 0, c'' \geq 0$, then the first-order approach is valid if the following conditions hold:*

- $G(\hat{R}, e) = \int_{-\infty}^{\hat{R}} F(R|e) \mathrm{d}R$ is non-increasing convex in $a$ for each value of $\hat{R}$, i.e., $\frac{\partial G(\hat{R}, e)}{\partial e} \leq 0, \frac{\partial^2 G(\hat{R}, e)}{\partial e^2} \geq 0$;
- $\mathbb{E}[R|e] = \int_{\mathcal{R}} R \cdot f(R|e) \mathrm{d}R$ is non-decreasing concave in $e$, i.e., $\frac{\partial \mathbb{E}[R|e]}{\partial e} \geq 0, \frac{\partial^2 \mathbb{E}[R|e]}{\partial e^2} \leq 0$;
- $MLRP: L(R|e) = \frac{f_e(R|e)}{f(R|e)}$ is non-decreasing concave in $R$ for each value of $e$, i.e., $\frac{\partial L(R|e)}{\partial R} \geq 0, \frac{\partial^2 L(R|e)}{\partial R^2} \leq 0$;
- $w(z) = u\left(u'^{-1}\left(1/z\right)\right)$ is concave.

Based on Lemma 1, let the user's payoff conditional on $R$ be $\pi(R)$. The incentive compatibility constraint then relaxes to:

$$\int_{\mathcal{R}} u(\pi(R)) \cdot f_e(R|e) \mathrm{d}R - c'(e) = 0. \tag{27}$$

where $f_e(R|e)$ is the derivative of $f(R|e)$ with respect to $e$.

## C Simulation Details

In this part of the Appendix, we provide the simulation setup details omitted from the main text and show additional numerical results.

## C.1 Simulation Detailed Setup

Assume the random factor $\tilde{y}$ follows an exponential distribution with parameter $\theta$:

$$f(y) = \frac{1}{\theta} \exp\left(-\frac{y}{\theta}\right). \tag{28}$$

The probability density function of uncertain privacy leakage $R$ is then given by:

$$f(R|e) = \frac{2}{\delta \theta e} \exp\left(\frac{2R}{\delta} - \frac{\exp\left(\frac{2R}{\delta}\right) - 1}{e\theta}\right), \tag{29}$$

and

$$f_e(R \mid e) = \frac{\exp\left(\frac{2R}{\delta}\right) - e\theta - 1}{e^2 \theta} \cdot f(R|e). \tag{30}$$

The cumulative distribution function is:

$$F(R|e) = 1 - \exp\left(\frac{1 - \exp\left(\frac{2R}{\delta}\right)}{e\theta}\right). \tag{31}$$

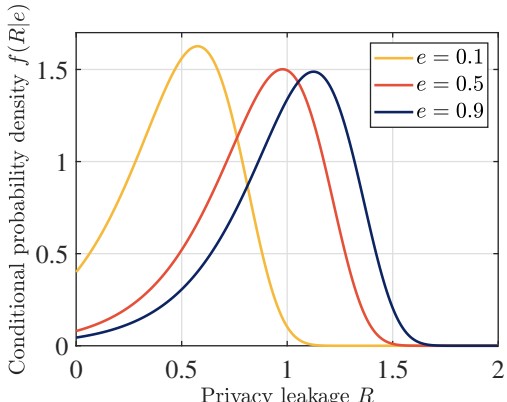

**Figure 10: Probability density function of privacy leakage $R$ conditional on user's effort $e$ when the random factor follows an exponential distribution with $\theta = 100$ and $\delta = 0.5$.**

We consider a constant $\delta = 0.5$, which is publicly known to both the platform and the user. This may be because the user must agree upon a data use agreement before they participate in the federated learning platform. Figure 10 illustrates how a user's effort affects the distribution of privacy risk. The utility function is defined as $u(\pi) = 1 - \exp(-r\pi)$, where $r$ represents the user's degree of risk aversion. Then using this setup, we can validate the first-order approach as described in Lemma 1.

Additionally, in our training cost setup for the user, it is important to note that $\lim_{e \to 1} c(e) = +\infty$, as $e = 1$ represents a perfectly trained model, which is impractical to achieve. Additionally, the user's utility function has an upper bound of 1, meaning that sufficiently high effort levels would render the incentive problem infeasible. To address this, we introduce a coefficient $\xi = 0.01$ to make the incentive problem feasible at higher effort levels, thus providing more valuable insights. We consider the effort range of $[0.05, 0.95]$. As $e \to 0$ will make the numerical instability due to the denominator in (29) goes to zero.

## C.2 Additionally Numerical Results

*C.2.1 Unlearning frequency.* For the platform's setting, we consider $S(R) = R^2$ and $q(R) = 0.1 \cdot R^2$. In this setting, $\mathcal{R}_U$ is given by $[0, \frac{p}{1.1}]$. For any realization of $R \in [0, \frac{p}{1.1}]$, the platform designs the compensation pairs $\hat{t}(R)$ and $t(R)$ to incentivize the user to request unlearning. The *unlearning frequency* (UF) is defined as:

$$\text{UF} = \int_{\mathcal{R}_U} f(R|e)\mathrm{d}R. \tag{32}$$

Figure 11 illustrates the unlearning frequency. As the user's privacy sensitivity increases, the range of $\mathcal{R}_U$ also expands. Additionally, as the user's effort increases, the distribution of $R$ shifts toward higher values (as shown in Figure 10), resulting in a decrease in unlearning frequency. This suggests a trade-off between incentive intensity and expected revenue in the incentive problem.

*C.2.2 An illustrative example for compensation scheme.* Figure 12 illustrates the user's payoff, with privacy sensitivity $p = 1.0$, as a function of the realized privacy leakage $R$, given that the user

exerts an effort level of $e = 0.85$. Then the contract terms of $C_E$ can de be determined according to Theorem 2. Additionally, unlearning imposes stricter limited liability, ensuring that the user's payoff in the unlearning-enabled platform remains positive. To incentivize the user to exert the effort the platform desires, the platform must offer a steep incentive. Moreover, when the user is more risk-averse, the difference in payoffs diminishes, as risk aversion also limits the unlearning-disabled platform's ability to offer high-intensity incentives.

*C.2.3 Platform's expected revenue and expected incentive cost.* Figure 13 shows the platform's expected revenue and the incentive costs required to induce a fixed effort level $e \in [0.05, 0.95]$. In both settings, the platform's incentive costs and expected surplus increase with the incentivized effort level $e$. As higher effort shifts the distribution of privacy leakage $R$ rightward. While the unlearning-disabled platform consistently generates higher expected revenue and incurs higher incentive costs, its expected surplus may be lower than that of the unlearning-enabled platform. The unlearning-disabled platform must induce effort within the range $e \in [0.735, 0.825]$ to maintain a positive surplus, whereas the unlearning-enabled platform can support effort over a larger range $e \in [0.435, 0.918]$.

## D Proofs

In this part of the Appendix, we provide the proofs omitted from the main text.

## D.1 Proof of Theorem 1

First, we show $\bar{t}_D$ and $\bar{t}_E$ are existing and unique. From the definition of $\bar{t}_D$,

$$\bar{t}_D = \inf \left\{ \bar{t} \geq 0 : \int_{\mathcal{R}} u\left(\bar{t} - pR\right) \cdot f(R|\bar{e})\mathrm{d}R - \bar{c} \geq 0 \right\}, \tag{33}$$

Since $u(\cdot)$ is an increasing function, the left-hand side of the inequality is increasing in $\bar{t}$. Thus, there is a unique value of $\bar{t}_D$ that satisfies

$$\int_0^{+\infty} u(\bar{t}_D - pR) \cdot f(R|\bar{e})\mathrm{d}R - \bar{c} = 0. \tag{34}$$

Similarly, $\bar{t}_E$ is defined as

$$\bar{t}_E = \inf \left\{ \bar{t} \geq 0 : \int_{\mathcal{R}} u\left(\max\{\bar{t} - pR, 0\}\right) \cdot f(R|\bar{e})\mathrm{d}R - \bar{c} \geq 0 \right\}. \tag{35}$$

For $\bar{t}_E$, the function $u\left(\max\{\bar{t} - pR, 0\}\right)$ behaves similarly to $u\left(\bar{t} - pR\right)$ except when $\bar{t} - pR \leq 0$, in which case it equals 0. Again, due to the monotonicity of $u(\cdot)$, there is a unique value of $\bar{t}_E$ that satisfies:

$$\int_0^{\frac{\bar{t}_E}{p}} u(\bar{t}_E - pR) \cdot f(R|\bar{e})\mathrm{d}R - \bar{c} = 0. \tag{36}$$

Thus, $\bar{t}_E$ is also existing and unique.

From the definitions of $\bar{t}_E$ and $\bar{t}_D$, we have the following relationship:

$$\int_0^{+\infty} u(\bar{t}_D - pR) \cdot f(R|\bar{e})\mathrm{d}R = \int_0^{\frac{\bar{t}_E}{p}} u(\bar{t}_E - pR) \cdot f(R|\bar{e})\mathrm{d}R. \tag{37}$$

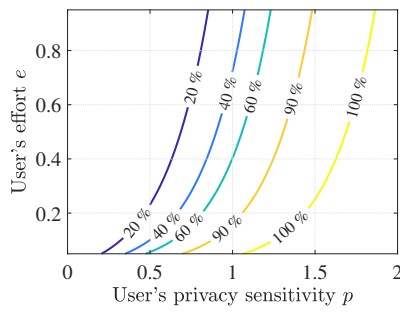

**Figure 11: Unlearning frequency versus user's privacy sensitivity $p$ and user's effort $e$.**

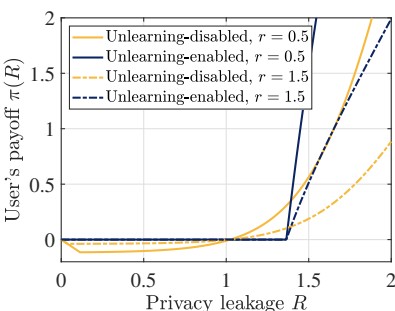

**Figure 12: Optimal contract to incentivize user's effort $e = 0.85$ when privacy sensitivity $p = 1.0$.**

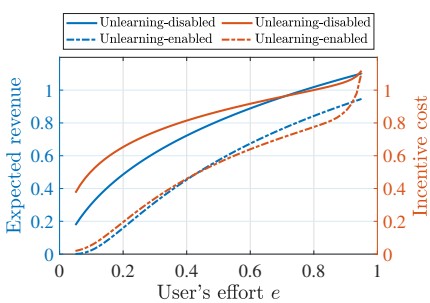

**Figure 13: Platform's expected revenue and incentive cost versus user's effort. The user risk aversion is $r = 0.5$ and privacy sensitivity is $p = 1.0$.**

Break down the integral on the left-hand side as:

$$\int_0^{+\infty} u(\bar{t}_D - pR) \cdot f(R|\bar{e}) \mathrm{d}R$$

$$= \int_0^{\frac{\bar{t}_E}{p}} u(\bar{t}_D - pR) \cdot f(R|\bar{e}) \mathrm{d}R + \int_{\frac{\bar{t}_E}{p}}^{+\infty} u(\bar{t}_D - pR) \cdot f(R|\bar{e}) \mathrm{d}R. \tag{38}$$

For the right-hand side, we have

$$\int_{\mathcal{R}} u(\max\{\bar{t}_E - pR, 0\}) \cdot f(R|\bar{e}) \, \mathrm{d}R = \int_0^{\frac{\bar{t}_E}{p}} u(\bar{t}_E - pR) \cdot f(R|\bar{e}) \, \mathrm{d}R. \tag{39}$$

Thus, the problem in (37) reduces to:

$$\int_0^{\frac{\bar{t}_E}{p}} [u(\bar{t}_E - pR) - u(\bar{t}_D - pR)] \cdot f(R|\bar{e}) \mathrm{d}R$$

$$= \int_{\frac{\bar{t}_E}{p}}^{+\infty} u(\bar{t}_D - pR) \cdot f(R|\bar{e}) \mathrm{d}R. \tag{40}$$

Using the concavity of $u(\cdot)$, we can bound the term $u(\bar{t}_D - pR)$ in right-hand side of (40):

$$u(\bar{t}_D - pR) \le u(\bar{t}_E - pR) + u'(\bar{t}_E - pR) \cdot (\bar{t}_D - \bar{t}_E), \tag{41}$$

Therefore, it follows that

$$\int_{\frac{\bar{t}_E}{p}}^{+\infty} u(\bar{t}_D - pR) \cdot f(R|\bar{e}) \mathrm{d}R$$

$$\le \int_{\frac{\bar{t}_E}{p}}^{+\infty} u(\bar{t}_E - pR) \cdot f(R|\bar{e}) \mathrm{d}R$$

$$\quad + (\bar{t}_D - \bar{t}_E) \cdot \int_{\frac{\bar{t}_E}{p}}^{+\infty} u'(\bar{t}_E - pR) \cdot f(R|\bar{e}) \mathrm{d}R. \tag{42}$$

For the left-hand side of (40),

$$\int_0^{\frac{\bar{t}_E}{p}} [u(\bar{t}_E - pR) - u(\bar{t}_D - pR)] \cdot f(R|\bar{e}) \mathrm{d}R$$

$$\ge (\bar{t}_E - \bar{t}_D) \cdot \int_0^{\frac{\bar{t}_E}{p}} u'(\bar{t}_E - pR) \cdot f(R|\bar{e}) \mathrm{d}R. \tag{43}$$

Finally, by combining the inequality (42) and (43), we obtain:

$$(\bar{t}_D - \bar{t}_E) \cdot \int_0^{+\infty} u'(\bar{t}_E - pR) \cdot f(R|\bar{e}) \mathrm{d}R$$

$$\ge - \int_{\frac{\bar{t}_E}{p}}^{+\infty} u(\bar{t}_E - pR) \cdot f(R|\bar{e}) \mathrm{d}R. \tag{44}$$

Rearranging the terms on both sides yields that:

$$(\bar{t}_D - \bar{t}_E) \ge \frac{- \int_{\frac{\bar{t}_E}{p}}^{+\infty} u(\bar{t}_E - pR) \cdot f(R|\bar{e}) \mathrm{d}R}{\int_0^{+\infty} u'(\bar{t}_E - pR) \cdot f(R|\bar{e}) \mathrm{d}R} \ge 0. \tag{45}$$

For $\frac{\bar{t}_D - \bar{t}_E}{\bar{t}_E}$, it can be bounded by:

$$\frac{\bar{t}_D - \bar{t}_E}{\bar{t}_E} \ge \frac{\bar{c} - \int_0^{+\infty} u(\bar{t}_E - pR) \cdot f(R|\bar{e}) \mathrm{d}R}{\bar{t}_E \cdot \int_0^{+\infty} u'(\bar{t}_E - pR) \cdot f(R|\bar{e}) \mathrm{d}R}$$

$$= \frac{\bar{c} - \mathbb{E}_R[u(\bar{t}_E - pR)]}{\bar{t}_E \cdot \mathbb{E}_R[u'(\bar{t}_E - pR)]}. \tag{46}$$

Using the Taylor expansion of $u(\bar{t}_E - pR)$ and $u'(\bar{t}_E - pR)$ around $\bar{t}_E$:

$$u(\bar{t}_E - pR) \approx u(\bar{t}_E) - pRu'(\bar{t}_E) + \frac{p^2 R^2}{2} u''(\bar{t}_E),$$
$$u'(\bar{t}_E - pR) \approx u'(\bar{t}_E) - pRu''(\bar{t}_E). \tag{47}$$

Substituting the definition of risk aversion $r = -\frac{u''(t)}{u'(t)}$ in (8) into both the numerator and the denominator of (46), we obtain:

$$\frac{\bar{t}_D - \bar{t}_E}{\bar{t}_E} \ge \frac{\bar{c} - u(\bar{t}_E) + p\mathbb{E}_R[R] \cdot u'(\bar{t}_E) + \frac{1}{2}p^2 \mathbb{E}_R[R^2] r \cdot u'(\bar{t}_E)}{\bar{t}_E u'(\bar{t}_E) \cdot (1 + pr\mathbb{E}_R[R])}$$

$$= \frac{\bar{c} - u(\bar{t}_E)}{\bar{t}_E u'(\bar{t}_E)} + \frac{p\mathbb{E}_R[R]}{\bar{t}_E} + \frac{1}{2} \frac{p^2 \mathbb{E}_R[R^2] r}{\bar{t}_E} \cdot \frac{1}{1 + pr\mathbb{E}_R[R]}. \tag{48}$$

Focus on the highest-order terms in $p$, we have

$$\frac{\bar{t}_D - \bar{t}_E}{\bar{t}_E} = \Omega(rp^2), \tag{49}$$

which shows that the relative difference between $\bar{t}_D$ and $\bar{t}_E$ grows quadratically with the $p$ and linearly with $r$.

## D.2 Proof of Theorem 2

We first present the contract design problem in (13) can be decomposed into two sub-problems: one for optimizing the user's utility extraction and another for determining the compensation terms $\hat{t}(R)$ and $t(R)$.

The platform's primal optimization problem in (13) can be expressed as:

$$\max_{e,a(R),\hat{t}(R),t(R)} \int_{\mathcal{R}} \big[ a(R) \cdot (-q(R) - \hat{t}(R))$$
$$+ (1 - a(R)) \cdot (S(R) - t(R)) \big] \cdot f(R|e) dR,$$

$$\text{s.t.} \int_{\mathcal{R}} u \big( a(R) \cdot \hat{t}(R)$$
$$+ (1 - a(R)) \cdot (t(R) - pR) \big) \cdot f(R|e) dR - c(e) \geq 0,$$

$$e \in \arg\max_{\hat{e} \in \mathcal{E}} \int_{\mathcal{R}} u \big( a(R) \cdot \hat{t}(R)$$
$$+ (1 - a(R)) \cdot (t(R) - pR) \big) \cdot f(R|e) dR - c(e),$$

$$a(R) \in \arg\max_{a \in \{0,1\}} a \cdot \hat{t}(R) + (1 - a) \cdot (t(R) - pR), \forall R \in \mathcal{R},$$

$$t(R) \geq 0, \hat{t}(R) \geq 0, \forall R \in \mathcal{R}.$$
$$(50)$$

Express the user's payoff conditional on $R$ as:

$$\pi(R) = a(R) \cdot \hat{t}(R) + (1 - a(R)) \cdot (t(R) - pR), \ \forall R \in \mathcal{R}. \quad (51)$$

The user's unlearning decision $a(R)$ is chosen to maximize its payoff under $R$, which implies:

$$a(R) = \mathbf{1}\left\{\hat{t}(R) \geq t(R) - pR\right\}, \ \forall R \in \mathcal{R}. \quad (52)$$

where $\mathbf{1}\{\cdot\}$ denotes the indicator function. Then the user's payoff $\pi(R)$ is determined by

$$\pi(R) = \max\left\{\hat{t}(R), t(R) - pR\right\}, \ \forall R \in \mathcal{R}. \quad (53)$$

Substituting the expression for $a(R)$ and $\pi(R)$ into (50), the platform's optimization problem can now be reformulated as:

$$\max_{\pi(R),t(R),\hat{t}(R)} \int_{\mathcal{R}} \big[ a(R) \cdot (-q(R))$$
$$+ (1 - a(R)) \cdot (S(R) - pR) \big] \cdot f(R|e) dR$$
$$- \int_{\mathcal{R}} \pi(R) \cdot f(R|e) dR, \quad (54a)$$

$$\text{s.t.} \int_{\mathcal{R}} u(\pi(R)) \cdot f(R|e) dR - c(e) \geq 0, \quad (54b)$$

$$e \in \arg\max_{\hat{e} \in \mathcal{E}} \int_{\mathcal{R}} u(\pi(R)) \cdot f(R|\hat{e}) dR - c(\hat{e}), \quad (54c)$$

$$\pi(R) = \max\left\{\hat{t}(R), t(R) - pR\right\}, \ \forall R \in \mathcal{R}, \quad (54d)$$

$$a(R) = \mathbf{1}\left\{\hat{t}(R) \geq t(R) - pR\right\}, \ \forall R \in \mathcal{R}, \quad (54e)$$

$$t(R) \geq 0, \hat{t}(R) \geq 0, \forall R \in \mathcal{R}. \quad (54f)$$

The problem naturally decomposes into two sub-problems. The first sub-problem focuses on determining the optimal payoff function $\pi(R)$, which satisfies the user's participation constraint (54b) and incentive compatibility constraint (54c). Once $\pi(R)$ is determined, the contract terms $\hat{t}(R)$ and $t(R)$ can be adjusted to maximize the platform's expected surplus (54a), ensuring that the chosen

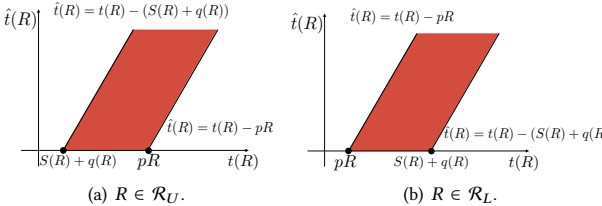

(a) $R \in \mathcal{R}_U$.

(b) $R \in \mathcal{R}_L$.

**Figure 14: Feasible contract region for each $R \in \mathcal{R}$. Given one contract term, the allowable range for another term can be determined within this region.**

compensation terms are consistent with the user's optimal payoff $\pi(R)$.

Based on the decomposition, the platform's adjusted revenue maximization is then expressed as:

$$\max_{t(R),\hat{t}(R)} \int_{\mathcal{R}} \big[ a(R) \cdot (-q(R))$$
$$+ (1 - a(R)) \cdot (S(R) - pR) \big] \cdot f(R|e) dR, \quad (55)$$

$$\text{s.t.} \quad a(R) = \mathbf{1}\left\{\hat{t}(R) \geq t(R) - pR\right\}, \ \forall R \in \mathcal{R}.$$

This adjusted revenue maximization problem simplifies the platform's objective by focusing on maximizing the expected surplus given the user's unlearning decision $a(R)$. By optimizing the objective function with respect to $a(R)$ for each $R \in \mathcal{R}$, we derive the necessary conditions for the adjusted revenue maximization in (55):

- If $-q(R) \geq S(R) - pR$, then the optimal user's unlearning decision for the platform is $a^*(R) = 1$, meaning that the contract must satisfy $t(R) - \hat{t}(R) \leq pR$.
- If $-q(R) < S(R) - pR$, then the optimal user's unlearning decision for the platform is $a^*(R) = 0$ and the contract must satisfy $t(R) - \hat{t}(R) \geq pR$.

Substituting the optimal $a^*(R)$ into platform's primal optimization in (50) yields a second set of necessary conditions:

- If $a^*(R) = 1$, the contract must also satisfy $t(R) - \hat{t}(R) \geq S(R) + q(R)$.
- If $a^*(R) = 0$, the contract must also satisfy $t(R) - \hat{t}(R) \leq S(R) + q(R)$.

Combining these conditions, we can state the Theorem 2.

We examine the necessity by driving a contradiction, i.e., any deviation from the contract structure specified in Theorem 2 results in a suboptimal expected surplus for the platform.

When $pR \geq S(R) + q(R)$, the optimal contract structure is:

$$S(R) + q(R) \leq t(R) - \hat{t}(R) \leq pR \quad (56)$$

Any deviation from this structure would imply:

- If the contract $\{\hat{t}(R), t(R)\}$ deviates such that $t(R) - \hat{t}(R) \leq S(R) + q(R) \leq pR$, the user will request unlearning to maximize its payoff under $R$. In this case, the platform's surplus becomes $-q(R) - \hat{t}(R)$, which is less than $S(R) - t(R)$, leading to a suboptimal expected surplus for the platform.

- Alternatively, if the contract deviates such that $S(R) + q(R) \leq pR \leq t(R) - \hat{t}(R)$, the user will not request unlearning because the condition $\hat{t}(R) \leq t(R) - pR$. The platform's surplus will then be $R - t(R)$, which is lower than $-q(R) - \hat{t}(R)$, again leading to suboptimality.

Similarly, when $pR \leq S(R) + q(R)$, any deviation from $pR \leq t(R) - \hat{t}(R) \leq S(R) + q(R)$ will result in a sub-optimal expected surplus for the platform.

Therefore, the contract structure specified in Theorem 2 is necessary. It ensures that the user selects the unlearning decision $a^*(R)$ that aligns with the platform's objective while simultaneously satisfying the user's IR and IC constraints.

We illustrate the relationship between $t(R)$ and $\hat{t}(R)$ for each $R \in \mathcal{R}$ in Figure 14.

## D.3 Proof of Theorem 3

In the binary privacy leakage case, the platform's revenues reduce to $\{S_H, S_L\}$, and the unlearning costs to $\{q_H, q_L\}$. The contract also reduce to $\bar{C}_D = \{t_H, t_L\}$ and $\bar{C}_E = \{t_H, t_L, \hat{t}_H, \hat{t}_L\}$.

When the unlearning is disabled, the contract design problem for binary privacy leakage is formulated as:

$$\max_{\bar{C}_D} F(e) \cdot (S_H - t_H) + (1 - F(e)) \cdot (S_L - t_L) \tag{57a}$$

$$\text{s.t. } F(e) \cdot u(t_H - pR_H) + (1 - F(e)) \cdot u(t_L - pR_L) - c(e) \geq 0, \tag{57b}$$

$$e \in \arg\max_{\bar{e} \in \mathcal{E}} F(\bar{e}) \cdot u(t_H - pR_H)$$
$$+ (1 - F(\bar{e})) \cdot u(t_L - pR_L) - c(\bar{e}), \tag{57c}$$

$$t_H \geq 0, t_L \geq 0. \tag{57d}$$

Replace the (57c) with its first-order condition based on Lemma 1. Then the optimization problem can be solved using the Lagrange method. Let $h(\cdot)$ denote the inverse function of $u(\cdot)$ with $h' > 0, h'' > 0$. The optimal contract is then given by:

- If $c(e) - \frac{F(e)}{F'(e)} \cdot c'(e) \geq u(-pR_L)$, the limited liability constraint (57d) is slack. In this case, $t_H = pR_H + h\left(c(e) + \frac{1 - F(e)}{F'(e)} \cdot c'(e)\right)$ and $t_L = pR_L + h\left(c(e) - \frac{F(e)}{F'(e)} \cdot c'(e)\right)$.
- If $c(e) - \frac{F(e)}{F'(e)} \cdot c'(e) < u(-pR_L)$, the limited liability constraint (57d) binds. Then $t_H = pR_H + h\left(u(-pR_L) + \frac{c'(e)}{F'(e)}\right)$ and $t_L = 0$.

Substituting these optimal contract terms into (57a) yields the unlearning-disabled platform's optimized expected surplus under effort $e$.

When the unlearning is enabled, the contract design problem becomes:

$$\max_{\bar{C}_E} F(e) \cdot \left[a_H \cdot (-q_H - \hat{t}_H) + (1 - a_H) \cdot (S_H - t_H)\right]$$
$$+ (1 - F(e)) \cdot \left[a_L \cdot (-q_L - \hat{t}_L) + (1 - a_L) \cdot (S_L - t_L)\right] \tag{58a}$$

$$\text{s.t. } F(e) \cdot u\left(a_H \cdot \hat{t}_H + (1 - a_H) \cdot (t_H - pR_H)\right)$$
$$+ (1 - F(e)) \cdot u\left(a_L \cdot \hat{t}_L + (1 - a_L) \cdot (t_L - pR_L)\right) - c(e) \geq 0, \tag{58b}$$

$$e \in \arg\max_{\bar{e} \in \mathcal{E}} F(\bar{e}) \cdot u\left(a_H \cdot \hat{t}_H + (1 - a_H) \cdot (t_H - pR_H)\right)$$
$$+ (1 - F(\bar{e})) \cdot u\left(a_L \cdot \hat{t}_L + (1 - a_L) \cdot (t_L - pR_L)\right) - c(\bar{e}), \tag{58c}$$

$$a_H = \arg\max_{a \in \{0,1\}} a \cdot \hat{t}_H + (1 - a) \cdot (t_H - pR_H), \tag{58d}$$

$$a_L = \arg\max_{a \in \{0,1\}} a \cdot \hat{t}_L + (1 - a) \cdot (t_L - pR_L), \tag{58e}$$

$$t_H \geq 0, t_L \geq 0, \hat{t}_H \geq 0, \hat{t}_L \geq 0. \tag{58f}$$

Considering the platform's strictly convex revenue function and unlearning cost function, we have the condition $\frac{S_L + q_L}{R_L} < \frac{S_H + q_H}{R_H}$. By applying Theorem 2, the contract design problem can be divided into three cases:

- $p \leq \frac{S_L + q_L}{R_L}$, where the platform designs $\bar{C}_E$ to induce $a_H = a_L = 0$.
- $\frac{S_L + q_L}{R_L} < p < \frac{S_H + q_H}{R_H}$, where the platform designs $\bar{C}_E$ to induce $a_H = 0, a_L = 1$.
- $p \geq \frac{S_H + q_H}{R_H}$, where the platform designs $\bar{C}_E$ to induce $a_H = 1, a_L = 1$.

Substitute (58c) with its first-order condition, and the optimal contract can be solved similarly to (57).

Comparing the surplus in both settings leads to three further cases:

**Case 1.** $c(e) - \frac{F(e)}{F'(e)} \cdot c'(e) \geq 0$, which implies $E_{\text{CR}}(e) \leq 1$ (high elasticity). The expected surplus difference is as follows:

(i) If $p \leq \frac{S_L + q_L}{R_L}$:

$$\mathbb{E}[\bar{V}_E] - \mathbb{E}[\bar{V}_D] = 0. \tag{59}$$

(ii) If $\frac{S_L + q_L}{R_L} < p < \frac{S_H + q_H}{R_H}$:

$$\mathbb{E}[\bar{V}_E] - \mathbb{E}[\bar{V}_D] = (1 - F(e)) \cdot R_L \left(p - \frac{S_L + q_L}{R_L}\right). \tag{60}$$

(iii) If $p \geq \frac{S_H + q_H}{R_H}$:

$$\mathbb{E}[\bar{V}_E] - \mathbb{E}[\bar{V}_D] = F(e) \cdot R_H (p - \frac{S_H + q_H}{R_H})$$
$$+ (1 - F(e)) \cdot R_L \left(p - \frac{S_L + q_L}{R_L}\right)$$
$$+ F(e) \cdot \left[h\left(c(e) + \frac{1 - F(e)}{F'(e)} c'(e)\right) - h\left(\frac{c'(e)}{F'(e)}\right)\right]$$
$$+ (1 - F(e)) \cdot h\left(c(e) - \frac{F(e)}{F'(e)} c'(e)\right). \tag{61}$$

In this case, the limited liability constraints (57d) and (58f) are slack.

When $p \leq \frac{S_L + q_L}{R_L}$, the unlearning-enabled platform induces the user not to unlearn regardless of the realization of privacy leakage. Then the two contract design problems are the same, resulting in zero surplus difference. When $\frac{S_L + q_L}{R_L} < p < \frac{S_H + q_H}{R_H}$, the unlearning-enabled platform induces the user to unlearn if privacy leakage $R_L$ is realized. At this point, the unlearning-enabled platform's surplus equals $-q_L - \hat{t}_L$ (with $\hat{t}_L = h\left(c(e) - \frac{F(e)}{F'(e)}c'(e)\right)$), which remains constant. Conversely, the unlearning-disabled platform absorbs a significant negative surplus when realized $R_L$, as the surplus is given by $S_L - pR_L - h\left(c(e) - \frac{F(e)}{F'(e)}c'(e)\right)$, which is linearly decreasing with $p$. When $p \geq \frac{S_H + q_H}{R_H}$, the unlearning-disabled platform incurs additional negative surplus when $R_H$ is realized.

**Case 2.** $u(-pR_L) < c(e) - \frac{F(e)}{F'(e)} \cdot c'(e) < 0$, which implies $1 < E_{CR}(e) < 1 - \frac{u(-pR_L)}{c(e)}$ (mediate elasticity). The expected surplus difference is as follows:

(i) $p \leq \frac{S_L + q_L}{R_L}$:

$$
\begin{aligned}
\mathbb{E}[\bar{V}_E] - \mathbb{E}[\bar{V}_D] = F(e) \cdot &\left[h\left(c(e) + \frac{1 - F(e)}{F'(e)}c'(e)\right) - h\left(\frac{c'(e)}{F'(e)}\right)\right] \\
&+ (1 - F(e)) \cdot h\left(c(e) - \frac{F(e)}{F'(e)}c'(e)\right).
\end{aligned}
\tag{62}
$$

(ii) $\frac{S_L + q_L}{R_L} < p < \frac{S_H + q_H}{R_H}$:

$$
\begin{aligned}
\mathbb{E}[\bar{V}_E] - \mathbb{E}[\bar{V}_D] = (1 - F(e)) \cdot R_L &\left(p - \frac{S_L + q_L}{R_L}\right) \\
+ F(e) \cdot &\left[h\left(c(e) + \frac{1 - F(e)}{F'(e)}c'(e)\right) - h\left(\frac{c'(e)}{F'(e)}\right)\right] \\
+ (1 - F(e)) \cdot &h\left(c(e) - \frac{F(e)}{F'(e)}c'(e)\right).
\end{aligned}
\tag{63}
$$

(iii) $p \geq \frac{S_H + q_H}{R_H}$:

$$
\begin{aligned}
\mathbb{E}[\bar{V}_E] - \mathbb{E}[\bar{V}_D] = F(e) \cdot R_H &\left(p - \frac{S_H + q_H}{R_H}\right) \\
+ (1 - F(e)) \cdot R_L &\left(p - \frac{S_L + q_L}{R_L}\right) \\
+ F(e) \cdot &\left[h\left(c(e) + \frac{1 - F(e)}{F'(e)}c'(e)\right) - h\left(\frac{c'(e)}{F'(e)}\right)\right] \\
+ (1 - F(e)) \cdot &h\left(c(e) - \frac{F(e)}{F'(e)}c'(e)\right).
\end{aligned}
\tag{64}
$$

In this case, the limited liability constraint (57d) in the unlearning-disabled platform's contract design remains slack, while in the unlearning-enabled platform, the limited liability constraint (58f) is binding and the individual rationality (IR) constraint (58b) is slack. This leads to a situation where the user's expected utility is greater than zero, creating a "limited liability rent". For sufficiently small values of $p$, where $\mathbb{E}[\bar{V}_E] - \mathbb{E}[\bar{V}_D] < 0$, the unlearning-enabled platform's surplus is dominated by this limited liability rent, resulting in a negative surplus difference.

**Case 3.** $c(e) - \frac{F(e)}{F'(e)} \cdot c'(e) \leq u(-pR_L)$, which implies $E_{CR}(e) \geq 1 - \frac{u(-pR_L)}{c(e)}$ (low elasticity). The expected surplus difference is as follows:

(i) $p \leq \frac{S_L + q_L}{R_L}$:

$$
\begin{aligned}
\mathbb{E}[\bar{V}_E] - \mathbb{E}[\bar{V}_D] = F(e) \cdot &\left[h\left(u(-pR_L) + \frac{c'(e)}{F'(e)}\right) - h\left(\frac{c'(e)}{F'(e)}\right)\right] \\
&- (1 - F(e)) \cdot pR_L
\end{aligned}
\tag{65}
$$

(ii) $\frac{S_L + q_L}{R_L} < p < \frac{S_H + q_H}{R_H}$:

$$
\begin{aligned}
\mathbb{E}[\bar{V}_E] - \mathbb{E}[\bar{V}_D] = F(e) \cdot &\left[h\left(u(-pR_L) + \frac{c'(e)}{F'(e)}\right) - h\left(\frac{c'(e)}{F'(e)}\right)\right] \\
&+ (1 - F(e))(-q_L - S_l).
\end{aligned}
\tag{66}
$$

(iii) $p \geq \frac{S_H + q_H}{R_H}$:

$$
\begin{aligned}
\mathbb{E}[\bar{V}_E] - \mathbb{E}[\bar{V}_D] = F(e) \cdot R_H &\left(p - \frac{S_H + q_H}{R_H}\right) \\
+ F(e) \cdot &\left[h\left(u(-pR_L) + \frac{c'(e)}{F'(e)}\right) - h\left(\frac{c'(e)}{F'(e)}\right)\right] \\
&+ (1 - F(e))(-q_L - S_L).
\end{aligned}
\tag{67}
$$

In this scenario, the limited liability constraints (57d) and (58f) are binding in both the unlearning-disabled and unlearning-enabled platforms, while the IR constraints (57b) and (57b) are slack. Both platforms must pay the user limited liability rent. In the unlearning-disabled setting, the user's utility is capped at $u(-pR_L)$, whereas in the unlearning-enabled setting, it is capped at zero (as shown in Figure 14). The unlearning-enabled platform must pay a higher limited liability rent, which increases with $p$, leading to a decreasing surplus difference.

