# OpenReview forum: "Unlearning Incentivizes Learning under Privacy Risk"
_ACM.org/TheWebConf/2025/Conference — WWW 2025 Oral_

### Official Review · Reviewer_KEXL · 2024-11-25

**Novelty:** 5
**Technical Quality:** 4

**Review:**

The authors are investigating how user's opportunity of machine unlearning affects both user behavior and platform profitability, specifically considering the scenarios where users can or cannot request data deletion from platforms.

User behavior is modeled by two key factors: their training effort level (which influences the associated cost) and privacy sensitivity (reflecting their attitude towards privacy loss). Without sufficient compensation from platform, users may lack motivation to participate in the learning process. So, the platform are described by two functions: revenue generated from user data exploitation and compensation costs paid to users for using their data. Using this framework, the authors define utility functions for both users and platforms under two scenarios: unlearning-enabled and unlearning-disabled conditions. They then formulate an abstract optimization problem to find the optimal contract that maximizes platform profit. For special cases that may arise in practice, the authors derive a set of theorems and provide detailed analysis for an illustrative example. Their findings show that users' privacy sensitivity has significant incentive effects, and in some instances, enabling unlearning can actually lead to greater profitability than disabling it.The authors also conduct a questionnaire survey, which reveals that enabling machine learning increases users' willingness to participate in federated learning.

While the perform numerical studies illustrate the incentive effect of unlearning and the working of the approach, translation of abstract concepts and functions to practical applications in real-world settings remains unclear.

In general, the current research in federated unlearning has mainly concentrated on developing effective and efficient unlearning techniques. The issue of incentivizing users to retain their data (this problem is considered by the authors) is still under-explored and makes it a pressing area of study, particularly given its increasing relevance to the development of intelligent services enabled by machine learning. Nevertheless, there is a study [Ding N. et al. Incentivized Federated Learning and Unlearning //arXiv preprint arXiv:2308.12502. – 2023.] that deserves to be mentioned in the article. They also investigate whether allowing federated unlearning is beneficial to the server and users, compared to a scenario without unlearning.

It seems that the work is closer to the track "Security and privacy".

**Questions:**

In general, the current research in federated unlearning has mainly concentrated on developing effective and efficient unlearning techniques. The issue of incentivizing users to retain their data (this problem is considered by the authors) is still under-explored and makes it a pressing area of study, particularly given its increasing relevance to the development of intelligent services enabled by machine learning. Nevertheless, there is a study [Ding N. et al. Incentivized Federated Learning and Unlearning //arXiv preprint arXiv:2308.12502. – 2023.] that deserves to be mentioned in the article.

It seems that the work is closer to the track "Security and privacy".

**Reviewer Confidence:**

3: The reviewer is confident but not certain that the evaluation is correct

**Scope:**

3: The work is somewhat relevant to the Web and to the track, and is of narrow interest to a sub-community

---

### Official Review · Reviewer_eELk · 2024-11-28

**Novelty:** 4
**Technical Quality:** 3

**Review:**

This paper addresses the economic impact of machine unlearning on user behavior and platform profitability. The authors begin by conducting a questionnaire survey, which demonstrates that machine unlearning enhances users’ willingness to participate in federated learning. They then establish a necessary condition for maximizing the surplus of an unlearning-enabled platform. Overall, the paper is well-structured, relatively clear, and demonstrates a certain level of innovation.

Pros:

1.	The paper explores several intriguing questions, such as: How does federated unlearning affect privacy-sensitive users’ willingness to participate in federated learning? How can one design an optimal contract that simultaneously incentivizes user participation in learning and appropriately elicits unlearning decisions? And how do users’ risk aversion and privacy sensitivity impact the platform’s profitability when unlearning is an available option? These research questions are thought-provoking and provide valuable inspiration for future work.
2.	The paper formally models and addresses these research problems in mathematical terms, offering theoretical support. Additionally, the authors complement their theoretical findings with insights from a questionnaire survey.
3.	The paper includes numerical experiments that yield interesting results. For instance, the numerical evaluation reveals that the platform’s profitability is heavily influenced by users’ privacy sensitivity. Notably, when users exhibit high levels of privacy sensitivity, enabling unlearning can significantly enhance profitability.

Cons:

1. The discussion of related works is insufficient. While there are numerous studies on incentive mechanisms in federated learning, the article lacks a thorough analysis and comparison of these works. This is a significant omission, as such studies are highly relevant to the research questions addressed in the paper. A more comprehensive review of related literature would strengthen the paper’s foundation and contextualize its contributions.
2. The experimental section is a weakness of the article. Relying solely on parameter-based simulations is not sufficiently convincing. The study would benefit greatly from validation using real datasets or the construction of realistic federated learning scenarios. While mathematical modeling can certainly be simulated, the data distributions of real users may differ significantly from those assumed in the simulations.
3. The definition of the research scope needs further clarification. While the article is clearly focused on federated learning, the abstract and parts of the text frequently use the broader term "machine learning" before switching to "federated learning." This inconsistency could confuse readers. Additionally, federated learning can be categorized into horizontal and vertical federated learning, each with distinct incentive and privacy challenges. Clearly defining the specific scenario under discussion would improve the paper’s focus and precision.

**Questions:**

1. The discussion of related works is insufficient. While there are numerous studies on incentive mechanisms in federated learning, the article lacks a thorough analysis and comparison of these works. This is a significant omission, as such studies are highly relevant to the research questions addressed in the paper. A more comprehensive review of related literature would strengthen the paper’s foundation and contextualize its contributions.
2. The definition of the research scope needs further clarification. While the article is clearly focused on federated learning, the abstract and parts of the text frequently use the broader term "machine learning" before switching to "federated learning." This inconsistency could confuse readers. Additionally, federated learning can be categorized into horizontal and vertical federated learning, each with distinct incentive and privacy challenges. Clearly defining the specific scenario under discussion would improve the paper’s focus and precision.

**Reviewer Confidence:**

3: The reviewer is confident but not certain that the evaluation is correct

**Scope:**

3: The work is somewhat relevant to the Web and to the track, and is of narrow interest to a sub-community

---

### Official Review · Reviewer_oQ2w · 2024-11-28

**Novelty:** 5
**Technical Quality:** 4

**Review:**

The paper examines the economic impacts of machine unlearning, a process designed to remove specific user data from a trained model to address privacy concerns. It formulates and analyzes a contract design framework for platforms operating in both unlearning-disabled and unlearning-enabled scenarios. Key challenges include balancing compensation for users’ learning and unlearning efforts, as well as managing privacy leakage and operational costs. A survey conducted reveals that enabling unlearning increases user participation in federated learning. The analysis shows that unlearning incentives grow quadratically with privacy sensitivity and suggests that platforms might profit more with unlearning under certain conditions.

The paper introduces a novel and timely framework addressing the interplay of privacy, economics, and user incentives in federated learning. The results offer actionable strategies for platforms to optimize contracts and balance costs, contributing to both theory and practice. The study is technically interesting. The overall presentation is good and I enjoy reading this paper.

However, Section 3.1 does not seem closely related to the main results; it would be better to shorten it or move it to the appendix. Additionally, the meaning of Theorem 1 is unclear, as it only states that the optimal compensation decreases without discussing the platform's surplus. Some key results, such as those in Sections 4.1 and 4.2.2, are limited to the binary scenario, leaving the structure of the optimal mechanism unclear.

A minor point: in Definition 2, are the terms on both sides of (7) reversed?

**Questions:**

1. Could you provide insights on when to choose unlearning-disabled versus unlearning-enabled scenarios with respect to r and p?
2. Could you elaborate on what the optimal incentivized effort e might look like?
3. How does the distribution of R influence the design of the optimal mechanism?
4. Do you have any thoughts on whether the results in Sections 4.1 and 4.2.2 can be generalized to broader cases?

**Reviewer Confidence:**

3: The reviewer is confident but not certain that the evaluation is correct

**Scope:**

3: The work is somewhat relevant to the Web and to the track, and is of narrow interest to a sub-community

---

### Official Review · Reviewer_qit8 · 2024-12-02

**Novelty:** 4
**Technical Quality:** 5

**Review:**

This paper investigates the economic impact of machine unlearning on user behavior and platform profitability. It formulates a series of contract design problems under scenarios with and without unlearning capabilities, analyzing the conditions that minimize incentive costs and maximize platform surplus. The findings suggest that enabling machine unlearning could, under certain conditions, lead to greater profitability compared to disabling it. Additionally, the authors conducted a questionnaire survey, revealing that machine unlearning enhances users’ willingness to engage in federated learning. Although realistic datasets were unavailable, the study relies on simulation experiments to support its conclusions.
Strengths:
1. This paper is well-constructed, with lucidity and straightforwardness.
2. This paper made simulation experiments, help clarify and understand proposed conclusions.
3. This paper proposes meaningful questions and derives reasonable conclusions which are valuable for platforms considering the deployment of machine unlearning.
Weaknesses:
1. Proof of Theorem 1 lacks additional assumption: “Since $u(\cdot)$ is an increasing function, the left-hand side of the inequality is increasing in $\overline{t}$. Thus, there is a unique value of $\overline{t}_D$ that satisfies...” and “Again, due to the monotonicity of $u(\cdot)$, there is a unique value of $\overline{t}_E$  that satisfies...”. However, this is not necessarily true for bounded utility functions, which differ from unbounded ones in this context. And in your simulation,  the utility function is defined as
$u(\pi) = 1-exp(−r\pi)$,  which has an upper bound of 1. In this situation, additional constraint on $\overline{c}$ is required to ensure the problem is solvable.
2. There are noticeable formula errors, though they do not affect the overall understanding, they should be corrected: In the proof of Theorem 1, in (38), (42) $\frac{t_E}{p}$ lacks superscript.

**Questions:**

See weaknesses.
Additional questions:
1. Is the assumption of a bounded utility function a fundamental premise of the study? Otherwise please claim it.
2. Will bounded utility function and unbounded utility function make a difference in simulation?

**Reviewer Confidence:**

3: The reviewer is confident but not certain that the evaluation is correct

**Scope:**

3: The work is somewhat relevant to the Web and to the track, and is of narrow interest to a sub-community

---

### Official Review · Reviewer_uwEa · 2024-12-02

**Novelty:** 5
**Technical Quality:** 5

**Review:**

The paper studies the impact of unlearning in incentivizing on users’ data-sharing behavior and the increasing cost due to decreased model performance. The authors formulated a set of research questions and provided their answers through a questionnaire, contract design, and numerical analysis.

The problem is clear and it seems a convincing gap in the literature. The authors did a good job introducing and formulating the problem, although it can help to have a notation table. What I found most interesting are the insights from the analysis, e.g., the positive effect of unlearning in more privacy-sensitive scenarios.

It would be useful to include more discussions on the implications of the findings.

I wonder if enabling unlearning will worsen privacy issues: if users are more willing to share their data given the unlearning option, does it eventually lead to more privacy issues?

**Questions:**

Implications of the findings.

**Reviewer Confidence:**

2: The reviewer is willing to defend the evaluation, but it is likely that the reviewer did not understand parts of the paper

**Scope:**

3: The work is somewhat relevant to the Web and to the track, and is of narrow interest to a sub-community